# An automated software-assisted approach for exploring metabolic susceptibility and degradation products in macromolecules using high-resolution mass spectrometry

Paula Cifuentes[1,2,3◔*], Ismael Zamora[3‡], Tatiana Radchenko[2‡], Fabien Fontaine[3‡], Albert Garriga[3‡], Luca Morettoni[4‡], Jesper Kammersgaard Christensen[5], Hans Helleberg[5], Bridget A. Becker[6]

**1** Pompeu Fabra University, Barcelona Spain, **2** Lead Molecular Design, S.L., Sant Cugat del Valles, Spain, **3** Mass Analytica, S.L., Sant Cugat Del Valles, Spain, **4** Mass Analytica, S.L., Bettona, Italy, **5** Development ADME, Novo Nordisk, Måløv, Denmark, **6** Labcorp, Madison, WI, United States of America

◔ These authors contributed equally to this work.
‡ These authors contributed equally to this work.
* paula.cifuentes01@estudiant.upf.edu

## Abstract

A comprehensive understanding of drug metabolism is crucial for advancements in drug development. Automation has improved various stages of this process, from compound procurement to data analysis, but significant challenges persist in the metabolite identification (MetID) of macromolecules due to their size, structural complexity, and associated computational demands. This study introduces new algorithms for automated Liquid Chromatography-High-Resolution Mass Spectrometry (LC-HRMS) data analysis applicable to macromolecules. A novel peak detection approach based on the most abundant mass (MaM) is presented and systematically compared with the monoisotopic mass (MiM) approach, commonly used in small molecules MetID. Additionally, three structure visualization strategies, expanded (atom-level), non-expanded (monomer-level), and a hybrid mode, are evaluated for their impact on computation data processing time and interpretability, based on their distinct fragmentation strategies. The workflow was validated using six diverse datasets, comprising linear and cyclic peptides and oligonucleotides with both natural and unnatural monomers, covering a molecular weight range of 700–7630 Da. A total of 970 metabolites were identified under various experimental and ionization conditions. The MaM algorithm demonstrated higher scores and a greater number of matches, instilling greater confidence in the accurate prediction of metabolite structures, while the non-expanded visualization significantly reduced processing times (ranging from minutes to under an hour for most peptides). Furthermore, the visualization algorithm, which integrates monomer-level and atom/bond notation, enables clear localization of metabolic biotransformations. Compared to previous studies, the proposed

**Data availability statement:** All relevant data are within the manuscript and its Supporting Information files.

**Funding:** This work has been partially supported by Doctorats Industrials, AGAUR, Generalitat de Catalunya. Industrial Doctorate grant 00002/2023.

**Competing interests:** P.C., F.F, and A.G are employees of Lead Molecular Design, S.L., and I.Z. is the CEO of the company. Lead Molecular Design, S.L. develops analytical software, including MassMetaSite and Oniro, which were used in this study. L.C is an employee for Mass Spec Analytica, S.L, the software distributor. Due to licensing restrictions, the source code of Oniro cannot be distributed directly. However, the software is accessible via a publicly available AWS-hosted server that includes a free trial option, allowing readers to use the software and reproduce the analyses described in the manuscript (https://oniro.mass-analytica.com/). This does not alter our adherence to PLOS ONE policies on sharing data and materials.

workflow demonstrated reduced processing time, consistent detection of degradation products, and enhanced visualization capabilities, advancing automated MetID for macromolecules.

## Introduction

An essential aspect of the drug development process is the comprehensive identification and characterization of the major metabolites of the drug candidate and the enzymes responsible for its metabolic transformation, commonly known as drug metabolism. These studies are crucial, as certain metabolites may exhibit superior potency or improved pharmacokinetics properties compared to the parent drugs, thereby enhancing therapeutic efficacy [1]. Conversely, some metabolites may be toxic or chemically reactive, potentially interfering with the metabolism of co-administered drugs and increasing the risk of drug-drug interactions [2, 3].

Therefore, MetID plays a vital role not only in guiding chemical modifications to improve metabolic stability and reduce toxicity, but also for informing clinical monitoring strategies and supporting personalized medicine approaches that aim to prevent adverse drug reactions. Collectively, these efforts are essential to the development of safe and effective therapeutic agents [4].

In recent years, the use of macromolecules such as peptides and oligonucleotides as therapeutic agents has rapidly grown in drug development, making MetID for these compounds increasingly important [5,6]. However, MetID for macromolecules presents greater challenges than for small molecules, especially in data analysis and result interpretation. The large size and structural complexity of such compounds, which often consists of hundreds of atoms, lead to an exponential increase in spectral signals that must be interpreted, along with a larger number of fragments to compute and compare, and difficulties to determine the specific location where the biotransformation has occurred [7]. Consequently, this complexity demands significantly more software processing time and memory.

Although several tools have been developed for automated MetID, these are primarily designed for small molecules and often struggle to process multiply charged states, which are prevalent in large biomolecules. Although some specialized approaches have been developed to address these challenges, they frequently suffer high false-positive rates in complex biological matrices and may offer limited support for various ionization modes, as highlighted in this research [7].

In our previous publications [8, 9] we developed software solutions focused on automating data analysis, primarily for small molecules, with some applicability to macromolecules. These tools have helped to create faster systems for the data processing step and the results review/visualization as they perform the following steps automatically: select the chromatographic peaks that are related to the compound of interest, find the mass spectral information for each extracted peak, assign potential structures by comparing the theoretical fragmentation that can be predicted with the actual mass to charge ratio (m/z) values obtained with the experimental spectra, scoring potential solutions depending on the fragments assigned to the spectra alone or by the comparison with the parent fragmentation. After clustering the results from

different experimental conditions and consolidating them into a single experimental entity, the results are stored in the database. Subsequently, upon the conclusion of the review process, a report is generated.

The primary aim of this article is to present novel algorithms and approaches for automated LC-HRMS data analysis that specifically address the challenges of MetID in macromolecules. One of the new approaches introduced in the automated workflow is a peak detection algorithm based on the MaM peak —an approach that, to our knowledge, has not been previously reported. To demonstrate its suitability for MetID of macromolecules, the algorithm is compared with the traditional MiM peak detection method, which has long been used in small molecule MetID studies.

In addition, two visualization strategies for macromolecules are presented. In the expand form all atoms and intermonomer bonds are shown, whereas in the non-expanded form, the structure is represented by linking the monomer acronyms. These visualizations have direct implications for the computational process: in the non-expanded form, the structure is not subjected to virtual atom-level metabolite generation. Instead, biotransformations are applied at the monomer level, which reduces the number of potential fragments generated and leads to decreased processing time and memory consumption. The non-expanded approach is compared to the expanded one, also demonstrating how this representation can facilitate the identification of biotransformation sites.

These proposed approaches are integrated into a workflow that enables the interpretation of data acquired under diverse experimental conditions and ionization modes. To validate the applicability of this workflow, analysis was conducted on six datasets spanning a molecular range from 700 to 7630 Da. These datasets consist of both linear and cyclic peptides, incorporating natural and unnatural amino acids, as well as oligonucleotides. Specifically, dataset-1 comprises 9 commercially available peptides, dataset-2 includes one commercially available peptide and 4 synthetic analogues, dataset-3 involves a natural peptide hormone and 7 synthetic analogues, dataset-4 features an antisense oligonucleotide, dataset-5 contains 28 commercially available peptides, and dataset-6 is composed of a peptide hormone. Covering macromolecules of varying sizes and structural types—including linear, cyclic, and non-standard monomers—these datasets demonstrate that the proposed methodology can be broadly applied across a wide compound applicability domain.

Comparisons of the results obtained for certain compounds with those of prior studies have enabled an evaluation of several factors, such as the number and structure of identified metabolites, along with a consideration of the time consumed during the data processing step.

## Materials and methods

### Experimental data

For this study, six different experimental data sets (linear/cyclic, natural/unnatural amino acids, and an oligonucleotide dataset) have been used for the MetID, as shown in Table 1. The proteases and biological matrices used in the experimental incubations of these datasets represent key relevant proteolytic environments that therapeutic peptides are likely to encounter in vivo. This includes enzymes involved in gastrointestinal metabolism—where peptide hydrolysis primarily occurs—such as trypsin, chymotrypsin, elastase, and pepsin. The other proteases and matrices reflect metabolism in the liver, blood, and other physiological contexts, ensuring coverage of a broad range of relevant peptide degradation pathways [10].

The first set (dataset-1) is composed of nine commercially available peptides (secretin, calcitonin, oxytocin, octreotide, deslorelin, histrelin, goserelin, buserelin, and leuprolide), each of them, was separately incubated, with four selected protease enzymes – trypsin, chymotrypsin, pancreatic elastase, and pepsin. Data acquisition was performed using a Thermo Orbitrap® instrument in full scan mode with data-dependent tandem mass spectrometry (MS/MS). The detailed experimental conditions for this dataset are documented in the referenced bibliography [11]. Three of the compounds are cyclic peptides (octreotide, oxytocin, and calcitonin) and five contain unnatural amino acids (secretin, calcitonin, ocreotide, deslorelin, and histrelin). Molecular weight ranges from 1282 to 3429 Da, as illustrated in Table 2.

Dataset-2 consists of a commercially available peptide glucagon-like peptide-1 (GLP-1), a 30 amino acid compound, and four synthetic analogues, designed to have a reduced susceptibility to enzymatic degradation, taspoglutide, exenatide, liraglutide and semaglutide, all of them linear peptides. MetID has been conducted under the presence of DPP-4

**Table 1. Summary of the number of compounds of each dataset, along with the molecular weight range of the compounds and the corresponding data acquisition mode. (DDA = data-dependent acquisition, DIA = data-independent acquisition).**

| Dataset | Number of compounds | Molecular weight range (Da) | Data acquisition mode | Incubation conditions |
|---|---|---|---|---|
| Dataset-1 | 9 | 1282–3429 | DDA | Trypsin, Chymotrypsin, Pancreatic Elastase, and Pepsin |
| Dataset-2 | 5 | 3298–4184 | DDA | Dipeptidyl peptidase-4 (DPP-4) and neutral endopeptidase (NEP) |
| Dataset-3 | 8 | 1637–1679 | DDA and DIA | Human Serum |
| Dataset-4 | 1 | 7633 | DDA | Human Liver |
| Dataset-5 | 25 | 708–1900 | DIA | Human Cathepsin G, Human Neutrophil Elastase, Human MMP-12 catalytic domain, and Bovine pancreatic trypsin |
| Dataset-6 | 1 | 5808 | DIA | Insulin-degrading enzyme (IDE) |

**Table 2. Dataset-1 sequence structures and its molecular weights.**

| Compound name | Molecular weight (Da) | Sequence | Structure |
|---|---|---|---|
| Deslorelin | 1282.45 | H-Pyr-His-Trp-Ser-Tyr-D-Trp-Leu-Arg-Pro-NHEt | Linear |
| Goserelin | 1269.41 | Glp-His-Trp-Ser-Tyr-Ser-tBu-Leu-Arg-Pro-NHNHCONH$_2$ | Linear |
| Buserelin | 1238.66 | Glp-His-Trp-Ser-Tyr-Ser-tBu-Leu-Arg-Pro-NHEt | Linear |
| Histrelin | 1323.5 | Glp-His-Trp-Ser-Tyr-HisBzl-Leu-Arg-Pro-NHEt | Linear |
| Leuprolide | 1209.4 | Glp-His-Trp-Ser-Tyr-D-Leu-Leu-Arg-Pro-NHEt | Linear |
| Secretin Human | 3039.41 | H-His-Ser-Asp-Gly-Thr-Phe-Thr-Ser-Glu-Leu-Ser-Arg-Leu-Arg-Glu-Gly-Ala-Arg-Leu-Gln-Arg-Leu-Leu-Gln-Gly-Leu-Val-NH$_2$ | Linear |
| Octreotide | 1019.24 | H-D-Phe-Cys (1)-Phe-D-Trp-Lys-Thr-Cys (1)-Thr-ol | Cyclic |
| Oxytocin | 1007.19 | H-Cys (1)-Tyr-Ile-Gln-Asn-Cys (1)-Pro-Leu-Gly-NH$_2$ | Cyclic |
| Calcitonin | 3429.71 | H-Cys (1)-Ser-Asn-Leu-Ser-Thr-Cys (1)-Val-Leu-Gly-Lys-Leu-Ser-Gln-Glu-Leu-His-Lys-Leu-Gln-Thr-Tyr-Pro-Arg-Thr-Asn-Thr-Gly-Ser-Gly-Thr-Pro-NH$_2$ | Cyclic |

and NEP, as both enzymes are known to be involved in native GLP-1 degradation. Data acquisition employed a Thermo Orbitrap® instrument operating in full scan mode with data-dependent MS/MS, as detailed previously in the cited references [11]. Except for semaglutide, which was incubated in dog plasma – with the two metabolites first synthesized and then spiked into the plasma – the data were collected using a Waters® ACQUITY® Ultra-Performance Liquid Chromatography with Vion Ion Mobility Spectrometry Quadrupole Time-of-Flight (IMS-QToF) Mass Spectrometer operated by UNIFI in a data-independent mode, in collaboration with Zealand Pharma. Taspoglutide peptide has non-natural amino acids and liraglutide has C-16 fatty acid side chain (palmitic acid). Molecular weights ranges from 3297 to 4184 Da, as presented in Table 3, being exenatide the larger.

Dataset-3 includes somatostatin, a natural growth-inhibiting peptide hormone, along with seven 14-amino acid cyclic analogues. Data is collected in two data acquisition modes; the first one was conducted on a Thermo Q-Exactive® instrument employing full scan mode with data-dependent MS/MS and the second one High Definition MS$^E$ (HDMS$^E$) data was collected using a Vion IMS QTof Mass Spectrometer. The detailed experimental conditions for this dataset are documented in the referenced bibliography [11, 12]. In the synthesis of these analogues, a common approach is employed, which entails substituting some of the natural amino acids with non-natural or modified ones (Fig 1). Notably, these analogues feature the substitution of Phe (7) by Msa, enhancing the rigidity due to the ortho substitution, and Trp (8) by D-Trp [13]. Additionally, various permutations involve substituting Ala (1), Cys (3), and Cys (14) with their D-amino acid equivalents, along with the substitution of Lys (4) by ornithine [13]. Molecular weight ranges from 1636 to 1678 Da Table 4. Given the inherent low stability of somatostatin, a critical consideration for its pharmaceutical utility, there is a great interest in evaluating whether these novel analogs (Table 4) exhibit prolonged lifetimes in human serum.

**Table 3. Dataset-2: sequence structures and molecular weights of GLP-1 and its analogues.**

| Compound name | Molecular weight (Da) | Sequence | Structure |
|---|---|---|---|
| GLP-1 | 3297.68 | H$_2$N-His-Ala-Glu-Gly-Thr-Phe-Thr-Ser-Asp-Val-Ser-Ser-Tyr-Leu-Glu-Gly-Gln-Ala-Ala-Lys-Glu-Phe-Ile-Ala-Trp-Leu-Val-Lys-Gly-Arg- Gly-OH | Linear |
| Liraglutide | 3751.20 | H-His-Ala-Glu-Gly-Thr-Phe-Thr-Ser-Asp-Val-Ser-Ser-Tyr-Leu-Glu-Gly-Gln-Ala-Ala-Lys(γ-Glu-palmitoyl)-Glu-Phe-Ile-Ala-Trp-Leu-Val-Arg-Gly-Arg-Gly-OH | Linear |
| Taspoglutide | 3338.71 | H-His-Aib-Glu-Gly-Thr-Phe-Thr-Ser-Asp-Val-Ser-Ser-Tyr-Leu-Glu-Gly-Gln-Ala-Ala-Lys-Glu-Phe-Ile-Ala-Trp-Leu-Val-Lys-Aib-Arg-NH$_2$ | Linear |
| Semaglutide | 4113.58 | H-His-Aib-Glu-Gly-Thr-Phe-Thr-Ser-Asp-Val-Ser-Ser-Tyr-Leu-Glu-Gly-Gln-Ala-Ala-Lys(γ-Glu-ADO-C$_{18}$ di-acid)-Glu-Phe-Ile-Ala-Trp-Leu-Val-Arg-Gly-Arg-Gly-OH | Linear |
| Exenatide | 4184.03 | H-His-Gly-Glu-Gly-Thr-Phe-Thr-Ser-Asp-Leu-Ser-Lys-Gln-Met-Glu-Glu-Glu-Ala-Val-Arg-Leu-Phe-Ile-Glu-Trp-Leu-Lys-Asn-Gly-Gly-Pro-Ser-Ser-Gly-Ala-Pro-Pro-Pro-Ser-NH$_2$ | Linear |

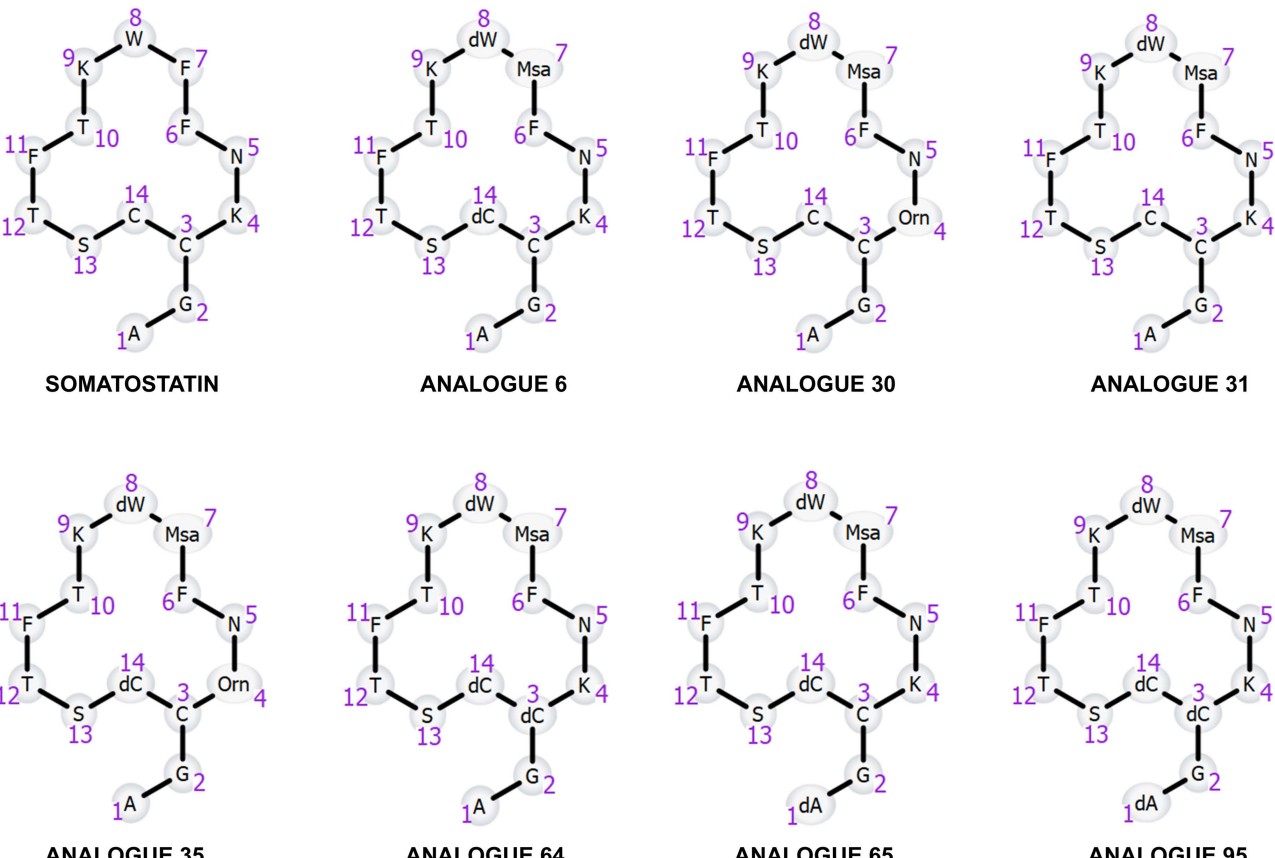

**Fig 1. Structure of somatostatin and its seven modified analogues including unnatural amino acids.** All eight peptides exhibit a cyclic structure, closing through the disulfide bond (between monomer 3 and 14).

Dataset-4 includes an antisense oligonucleotide (ASOs) with the formula C$_{242}$H$_{307}$N$_{91}$O$_{150}$P$_{94}$ (molecular weight of 7633 Da) containing 25 monomers. ASOs are synthetic, small-sized single-stranded nucleic acids. Data was collected using a Thermo Orbitrap® instrument in DDA mode. This dataset pertains to the incubation of ASOs in human liver tissue, a commonly studied experimental condition [14]. It enables researchers to evaluate the efficacy and selectivity of the ASOs in targeting specific messenger RNA molecules within the complex environment of the liver.

**Table 4. Dataset-3 is composed of somatostatin and its seven modified analogues, with the corresponding molecular formulas and molecular weights.**

| Compound name | Molecular formula | Monoisotopic mass (Da) | Structure |
|---|---|---|---|
| Somatostatin | $C_{76}H_{104}N_{18}O_{19}S_2$ | 1636.7167 | Cyclic |
| Analogue 6 | $C_{79}H_{110}N_{18}O_{19}S_2$ | 1678.7636 | Cyclic |
| Analogue 30 | $C_{78}H_{108}N_{18}O_{19}S_2$ | 1664.7480 | Cyclic |
| Analogue 31 | $C_{79}H_{110}N_{18}O_{19}S_2$ | 1678.7636 | Cyclic |
| Analogue 35 | $C_{78}H_{108}N_{18}O_{19}S_2$ | 1664.7480 | Cyclic |
| Analogue 64 | $C_{79}H_{110}N_{18}O_{19}S_2$ | 1678.7636 | Cyclic |
| Analogue 65 | $C_{79}H_{110}N_{18}O_{19}S_2$ | 1678.7636 | Cyclic |
| Analogue 95 | $C_{79}H_{110}N_{18}O_{19}S_2$ | 1678.7636 | Cyclic |

In this study, dataset-5 comprises a collection of 25 structurally diverse linear and cyclic peptides, with molecular weights ranging from 708 to 1900 Da (atosiban, BIO-11006, BIO-1211, carbetocin, CSP7, deslorelin, desmopressin, felypressin, gonadorelin, iseganan, lanreotide, LDTRYLEQLHKLY, leuprolide, lypressin, M10 peptide, MMI-0100, NAS-911, ocreotide, peptide T, salmon calcitonin, somatostatin, SPX-101, triptorelin, vasopressin, and vapreotide), as depicted in Table 5. These compounds have been incubated with four pulmonary proteases (human cathepsin G, human neutrophil elastase, human

**Table 5. Dataset-5, composed of 28 peptides, with the corresponding sequence structures and molecular weights.**

| Compound name | Molecular weight (Da) | Sequence | Structure |
|---|---|---|---|
| BIO-1211 | 708.8 | 4-[(2-tolyl)-urea]-phenylacetyl-Leu-Asp-Val-Pro-OH | Linear |
| CSP7 | 815.92 | H-Phe-Thr-Thr-Phe-Thr-Val-Thr-OH | Linear |
| Peptide T | 857.87 | H-Ala-Ser-Thr-Thr-Thr-Asn-Tyr-Thr-OH | Linear |
| BIO-11006 | 1050.18 | Ac-Gly-Ala-Gln-Phe-Ser-Lys-Thr-Ala-Ala-Lys-OH | Linear |
| SPX-101 | 1179.38 | H-D-Ala-D-Ala-Leu-Pro-Ile-Pro-Leu-Asp-Glu-Thr-D-Ala-D-Ala-OH | Linear |
| M10 peptide | 1181.27 | H-Thr-Arg-Pro-Ala-Ser-Phe-Trp-Glu-Thr-Ser-OH | Linear |
| Gonadorelin | 1182.31 | H-Pyr-His-Trp-Ser-Tyr-Gly-Leu-Arg-Pro-Gly-NH$_2$ | Linear |
| Leuprolide | 1209.42 | H-Pyr-His-Trp-Ser-Tyr-D-Leu-Leu-Arg-Pro-NHEt | Linear |
| Deslorelin | 1282.48 | H-Pyr-His-Trp-Ser-Tyr-D-Trp-Leu-Arg-Pro-NHEt | Linear |
| Triptorelin | 1311.47 | H-Pyr-His-Trp-Ser-Tyr-D-Trp-Leu-Arg-Pro-Gly-OH | Linear |
| NAS-911 | 1393.68 | H-Arg-Pro-Lys-Pro-Gln-Gln-Phe-Phe-Sar-Leu-Met(O2)-NH$_2$ | Linear |
| LDTRYLEQLHKLY | 1691.95 | H-Leu-Asp-Thr-Arg-Tyr-Leu-Glu-Gln-Leu-His-Lys-Leu-Tyr-OH | Linear |
| MMI-0100 | 2283.68 | H-Tyr-Ala-Arg-Ala-Ala-Ala-Arg-Gln-Ala-Arg-Ala-Lys-Ala-Leu-Ala-Arg-Gln-Leu-Gly-Val-Ala-Ala-OH | Linear |
| Salmon Calcitonin | 3431.89 | H-Cys (1)-Ser-Asn-Leu-Ser-Thr-Cys (1)-Val-Leu-Gly-Lys-Leu-Ser-Gln-Glu-Leu-His-Lys-Leu-Gln-Thr-Tyr-Pro-Arg-Thr-Asn-Thr-Gly-Ser-Gly-Thr-Pro-NH$_2$ | Linear |
| Carbetocin | 988.17 | deamino-Cys (1)-Tyr(Me)-Ile-Gln-Asn-Cys (1)-Pro-Leu-Gly-NH$_2$ | Cyclic |
| Atosiban | 994.19 | deamino-Cys (1)-D-Tyr(Et)-Ile-Thr-Asn-Cys (1)-Pro-Orn-Gly-NH$_2$ | Cyclic |
| Octreotide | 1019.25 | H-D-Phe-Cys (1)-Phe-D-Trp-Lys-Thr-Cys (1)-Thr-ol | Cyclic |
| Felypressin | 1040.23 | H-Cys (1)-Phe-Phe-Gln-Asn-Cys (1)-Pro-Lys-Gly-NH$_2$ | Cyclic |
| Lypressin | 1056.23 | H-Cys (1)-Tyr-Phe-Gln-Asn-Cys (1)-Pro-Lys-Gly-NH$_2$ | Cyclic |
| Desmopressin | 1069.22 | deamino-Cys (1)-Tyr-Phe-Gln-Asn-Cys (1)-Pro-D-Arg-Gly-NH$_2$ | Cyclic |
| Vasopressin | 1084.24 | H-Cys (1)-Tyr-Phe-Gln-Asn-Cys (1)-Pro-Arg-Gly-NH$_2$ | Cyclic |
| Lanreotide | 1096.33 | H-D-2Nal-Cys (1)-Tyr-D-Trp-Lys-Val-Cys (1)-Thr-NH$_2$ | Cyclic |
| Vapreotide | 1131.38 | H-D-Phe-Cys (1)-Tyr-D-Trp-Lys-Val-Cys (1)-Trp-NH$_2$ | Cyclic |
| Somatostatin | 1637.88 | H-Ala-Gly-Cys (1)-Lys-Asn-Phe-Phe-Trp-Lys-Thr-Phe-Thr-Ser-Cys (1)-OH | Cyclic |
| Iseganan | 1900.28 | H-Arg-Gly-Gly-Leu-Cys (1)-Tyr-Cys (2)-Arg-Gly-Arg-Phe-Cys (2)-Val-Cys (1)- Val-Gly-Arg-NH$_2$ | Cyclic |

**Fig 2. Insulin structure with the linear visualization.** The structure of insulin consists of two peptide chains known as Chain A, comprising 21 amino acids (numbered 1–21), and Chain B, comprising 30 amino acids (numbered 22–51). The A and B chains are interconnected by two disulfide bonds (highlighted in pink and light blue), and an additional disulfide bond is formed within the A Chain (highlighted in purple).

MMP-12 catalytic domain, and bovine pancreatic trypsin). Except felypressin, iseganan, LDTRYKEQLHKLY, lypressin, MMI-0100, vasopressin that data is unavailable for bovine pancreatic trypsin incubation, and atosiban, lanreotide, leuprolide which data is also unavailable for the human cathepsin G protease incubation. Data acquisition was performed using a Waters® Q-TOF instrument in a data-independent mode. The data was used to develop an assay workflow aimed at guiding the initial chemical modifications of peptide hits in early respiratory drug discovery projects. The detailed experimental conditions for this dataset are documented in the referenced bibliography [15]. This workflow utilizes WebMetabase to effectively detect and elucidate the structures of metabolites formed through enzymatic proteolysis. This data has been used in this study for a comprehensive comparison of results obtained through this new approach. Furthermore, its utilization serves to underscore the noteworthy advancements in data processing time realized through the implementation of this workflow.

Dataset-6 comprises human insulin, a peptide hormone containing three disulfide bridges, one of which is internally located within Chain A, while the other two covalently connect Chain A to Chain B (Fig 2). Data was collected with QTOF from a Waters® instrument. Insulin has been subjected to analysis following incubation with IDE, a protease widely recognized for its pivotal role in degrading and inactivating insulin. The detailed experimental conditions for this dataset are documented in the referenced bibliography [16].

## Data preprocessing

The MassMetaSite procedure consists of three steps: **(a) data reading, (b) automatic detection of the chromatographic peaks related to the parent compound and its metabolites,** and **(c) structure elucidation** by proposing a potential metabolite structure based on the fragmentation pattern for each peak detected in the previous step.

**a) Data reading.** Three different acquisition files need to be defined, depending on the data. Firstly, a blank file is employed to distinguish relevant signals from background peaks. This file is crucial for investigating whether a detected peak in the incubation file is attributable to the compound of interest or if it was already present in the incubation matrix (blank sample). Secondly, a substrate file is utilized to analyze the fragmentation pattern of the substrate. This step is essential in the structure elucidation process, involving the comparison of fragments assigned to the spectra of the parent compound with the spectra of potential metabolites. Lastly, the incubation file which contains all the products after incubation, either *in vitro* or *in vivo*. It serves for investigating and identifying metabolites formed during the incubation process.

**b) Automatic detection of the chromatographic peaks.** During the automated chromatographic peak detection stage, an initial spectral noise analysis is conducted. For each full scan (intensity vs. m/z), a noise level is computed by calculating the change in slope between two consecutive shortlists of ions present in the full scan, and ions below this threshold are systematically eliminated. Subsequently, the list of ions is examined across chromatographic retention times. Ions are selected based on specific m/z values to precisely determine the presence or absence of peak formation.

Following the identification of a potential peak in the incubation sample, a background analysis is performed. Specifically, for the selected m/z and retention time of the potential peak, a search is conducted to verify the presence of the

peak in the blank sample. If the peak is detected in the blank, a peak alignment optimization is initiated using a combination of Hodgkin and Pearson similarity indexes computation, which allows a comprehensive comparison of both shape and peak intensity. The sample peak is excluded from the analysis whenever it exhibits similar shape and equivalent (or lower) intensity to the blank peak. The Negative Control Area Ratio is then computed, representing the quantitative ratio between the peak area in the incubation sample and the corresponding in the blank.

Subsequently, a filtered spectrum is computed by merging all the scans within the peak retention time range. This involves the selection of m/z values that exhibit correlation within the chromatographic peak shape. Each m/z value of each filtered spectrum is compared with any of the m/z values for the metabolites of the parent compound. There are two potential options to represent the theoretical m/z of the compound of the peak under consideration: the monoisotopic or the most abundant isotope species. Additionally, the isotope pattern derived from the metabolite formula is compared to the one from the experimental spectra and a filter may be set to consider the similarity between the observed and predicted intensity for each potential isotope. In addition, m/z values from multiple charge states were also used in the analysis.

For each selected m/z value extracted from the filtered spectra, a comprehensive metabolite classification is conducted. This classification categorizes metabolites into distinct groups, including first-generation metabolites, second or higher generation metabolites, metabolites stemming from biotransformations unrecognized by the software (referred to as "red peaks" denoting unknowns), and cases where the fragment ion may arise from ion adduct formation or in-source neutral loss.

Ultimately, a MS/MS evaluation is conducted, examining the presence of m/z values observed in the parent spectrum within the potential peak. The evaluation considers the shift based on the obtained formula, classifying a non-shifted scenario when the same m/z observed in the parent spectra is also observed in the metabolite, and identifying a shift when a change in the m/z of the considered value relative to the parent is observed between a peak in the parent spectra and a peak in the filtered spectra.

The m/z values are scored according to multiple criteria: isotope similarity, retention time, MS/MS comparison and calculated m/z. Among all the values above the score threshold, the m/z that will represent the peak in the chromatogram is the one with the highest m/z value. This process results in a compiled list of peaks, each associated with an assigned m/z, retention time range, area, full scan filtered spectra, and MS/MS spectra.

**c) Structure elucidation**. The third stage of data processing is structure elucidation (Fig 3), during which the fragment ions obtained from the parent and those from the metabolite are compared.

This process has two starting points: the parent structure, or the metabolite structure which is obtained by virtual synthesis:

1. **Identification of metabolite fragments from fragmentation of the parent**

   1.1 Parent fragmentation: During this process, the parent molecule is fragmented, and the m/z of the fragments are computed. There could be more than one m/z value for a single fragment due to potential hydrogen rearrangements. Fragment structures are then associated to the spectra m/z values considering a user-specified tolerance.

   1.2 Generation of metabolite fragments: Metabolite fragments are built from parent fragments using metabolite and parent atom map. The metabolite resulting fragments m/z may be shifted or equal to the parent m/z depending on whether the fragment contains sites of metabolism or not [17].

   1.3 Association between parent peaks and metabolite peaks: For each parent spectrum, whether MS or MS/MS, the software checks if there are peaks with the same or shifted m/z in the associated metabolite spectrum. A shifted m/z is equal to the m/z of the parent plus the change of m/z due to the chemical modifications introduced during metabolism. Resulting in Substrate-Metabolite peak pairs that could be used for structural identification.

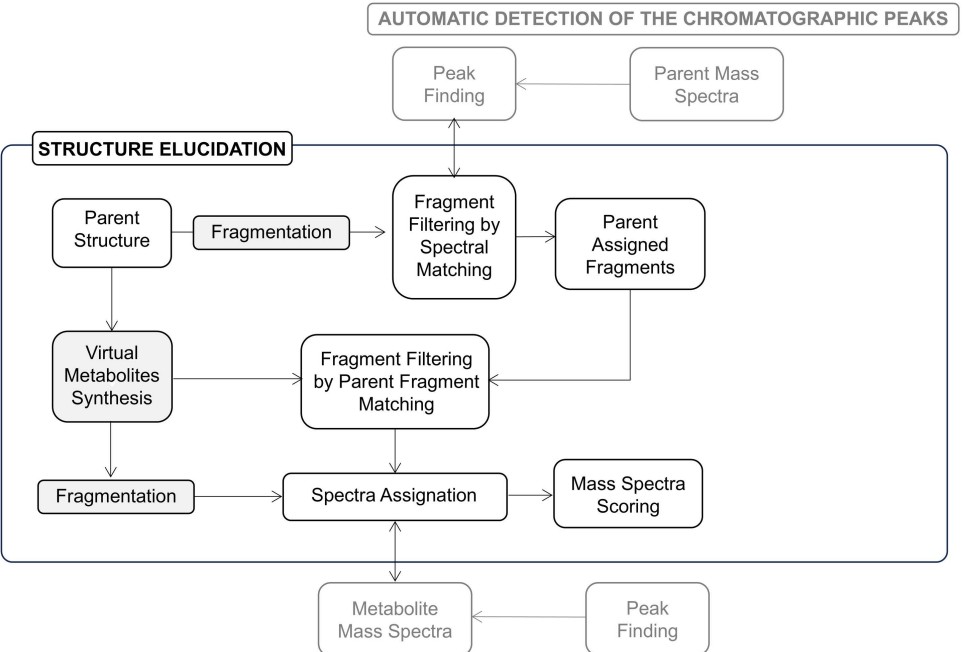

**Fig 3. Illustrates the third step, Structure Elucidation, of MassMetaSite procedure.**

a) Matches: When substrate and metabolite fragments are identical and both peaks of the Substrate-Metabolite fragment pair have the same m/z value, the observed and calculated interpretation match. Likewise, when the metabolite fragment is different from the substrate fragment and the Substrate-Metabolite fragment pair have a shifted mass, the interpretations also match [18].

b) Mismatches: The fragments that are mismatching are those ones where the m/z is observed as non-shifted between the parent and metabolite spectra, but the atom set of the fragment corresponds to a chemical modification that would change the m/z. Similarly, a mismatch is detected when the m/z is observed as shifted between the parent and the metabolite spectra, but the atom set of the fragment corresponds to a modification that would not change the m/z of the fragment [18].

2. **Identification of metabolite fragment from the structure of the metabolite:** Virtual fragments of the metabolites are generated based on a predefined list of metabolic biotransformation reactions [19].

a) Fragmentation of the metabolite: This is the same as the parent fragmentation but the number of bonds that can be cut is usually lower since breaking all the possible metabolites has a greater computational cost.

b) Metmatches: The fragments that are obtained in this way are assigned to the metabolite spectra are called metmatches. This fragmentation strategy is particularly beneficial for cyclic peptides, where the metabolite might be a linear peptide due to amide hydrolysis-induced ring opening, leading to a markedly different fragmentation pattern compared to the parent.

Scoring is done by summing the intensity for the matching peaks plus the sum of the intensity for the metmatching peaks minus the sum of the intensity for the mismatching peaks. The solutions with the highest score are auto selected by the system and reported as potential structural candidates [18].

Each experiment consisted of a set of samples, i.e., one sample per incubation time point per matrix. MassMetaSite processes each sample as a separate entity, and thus generates three main pieces of information for each sample: metabolic scheme, spectrometry data (product ion assignment) and outcomes (retention time, MS area, MS relative area, collision cross section, and parts per million (ppm) mass error) for each found component. WebMetabase then consolidates all these data from the individual files into a single interpretation for the entire experiment (time/matrix) and analyses which metabolite peaks from each sample can be clustered based on its retention time and m/z.

## Settings/Structure visualization

In this study, data have been processed with distinct algorithms, establishing the groundwork for a comprehensive comparison among them. This research is focused on three crucial dimensions

-**Peak detection (Monoisotopic Mass and Most Abundant Mass).** Various algorithms for peak detection are employed based on the molecular size. The MiM represents the peak to the ion with the lowest mass-to-charge (m/z) ratio and it is calculated using the lightest isotope mass of each element present in the molecule. It is particularly useful for accurately determining the molecular formula, especially for smaller molecules [20]. Conversely, MaM represents the molecule's most common isotopic distribution, considering the natural abundance of all isotopes in the molecule, not just the lightest ones.

For larger molecules or when the monoisotopic ion is undetectable, the MaM is employed for peak detection. This choice is made because, with increasing molecular size, the heightened probability of the entire molecule containing at least one heavy isotope atom (mainly $^{13}$C) becomes more pronounced. Consequently, the MiM peak may be much more difficult to detect than the MaM peak. In addition, MaM peaks are typically the ones which are selected for triggering MSMS scans in DDA when no preferred list is provided to the acquisition software.

In this study all datasets have been processed with both the MiM and MaM algorithms, except dataset-5 that has been exclusively subjected to processing with the MaM settings.

-**Acquisition modes (Data-dependent acquisition and Data-independent acquisition)** The LC-HRMS stands as the preferred method MetID, with DDA being commonly used strategy in MS data acquisition. In DDA, precursor ions selected based on their abundances are often employed to drive MS/MS. In contrast, DIA methods, such as MS$^E$ and HDMS$^E$, eliminate the risk of overlooking metabolites by avoiding precursor ion selection [21]. The DIA HDMS$^E$ is a method that combines ion mobility separation with MS$^E$ data acquisition. It alternates between low and high collision energy ion mobility spectrometry-mass spectrometry scans, enabling accurate mass measurements of both precursor and product ions simultaneously. In contrast to MiM, where a specific m/z must be isolated before fragmentation, DIA provides more complex but more complete datasets.

Data from dataset-3 was acquired employing the two predetermined strategies, DIA and DDA, facilitating a comparison of outcomes obtained from both acquisition modes. Settings used for the processing of DIA (MS$^E$/HDMS$^E$) and DDA data for somatostatin synthetic analogues are presented in S6-S9 Files.

-**Structure visualization (Expanded and non-expanded).** Two visualization options are available for representing the structure of polymeric compounds like peptides or oligonucleotides during data analysis. Monomers of the compound can be depicted either in an expanded form, revealing all atoms and intermonomer bonds, or in a non-expanded form, where the structure is represented by linking the monomer acronyms. In this study, dataset-4 was processed using both visualization options, enabling a comparative analysis of processing time and providing an illustrative example of how metabolites structures are visualized after metabolic reactions using both approaches.

The selection to work on expanded or non-expanded monomers has an impact on structure visualization. The non-expanded mode shows the monomer symbol making it simpler for the user to identify the structure and the place where the biotransformation takes place and therefore it is recommended to be used. Nevertheless, it also has implications in the computation process. The structure that is represented as monomer does not undergo a virtual structure

metabolite generation, the biotransformation is applied at monomer level and not at atomic level, therefore the resulting compound is not a valid chemical structure, since there is no information on the exact chemical structure that is obtained after the reaction. The part of the structure that is represented as atoms/bonds undergoes a typical virtual reaction and a defined chemical structure is obtained for each potential metabolite. In the monomer presented part of the molecule, fewer chemical structures need to be constructed during the calculation process, resulting in reduced computation time. There is another aspect applied on the part of all the molecule that is treated as monomer, since for this part only the typical a,b,c, and x,y,z fragmentation is considered, reducing in this way the number of potential fragments generated degreasing the time and memory consumption. For the rest of the molecule treated as atoms/bond all the bonds are disconnected to generate fragments that will generate an increased number of fragments.

Furthermore, there exists the option to work with a combination of both visualizations within the molecular structure. This can be achieved by selectively choosing which segments of the molecule to expand or maintain in a non-expanded state.

## Data analysis

Following data consolidation, manual data interpretation by the user is conducted for peak selection and structure elucidation steps, applying diverse data analysis criteria to systematically eliminate any potential false positive metabolites. These criteria are:

## Peak selection

- **MS area (%):** Reporting with a relative area above 0.5%.

- **Difference between observed and calculated m/z (amu, ppm):** For the MS signal the system computes the difference between the observed and the computed m/z. The observed m/z considers the m/z finds at the different scans and derives a value which is compared to the vendor software package to consider effects like peak saturation and loss of accuracy at the top of the peak. Maintaining a difference of less than 10 ppm between observed and computed values [22].

- **Value of isotopic all similarity:** Quantifying the match between observed and expected isotopic patterns for peaks, where a low value suggests pattern variability.

- **Negative control area ratio:** Establishing the ratio between peak areas in the incubation sample and the blank, with a signal observed in both considered non-specific.

- **Kinetics:** Reflects changes in metabolite abundance over time. At time 0 (t=0), when the incubation begins, the cluster chart would initially show the presence of ions solely related to the parent compound. There should be no signals corresponding to metabolites at this point, as no biotransformation has occurred yet. The first generated metabolite usually has an exponential shape, as they are starting to be formed. If the metabolites are further metabolized, the signal of the metabolite will decrease since the metabolite has been consumed to generate a second generation one. Typically, the second-generation metabolite has a sigmoidal shape since it needs the first-generation metabolite to form and then be further metabolized [11].

- **Shape of the metabolite peak:** Ideally, metabolite peaks should exhibit a Gaussian shape; however, in practice, peak tails may occasionally occur. It is important to distinguish these from peaks that resemble background noise or exhibit irregular shapes, such as broad or asymmetric profiles, which may suggest contamination or interference rather than the presence of a true metabolite [23].

**Structure elucidation.** The second step of the algorithm proposes potential metabolite structure based on the fragmentation pattern for each peak detected in the peak selection step Fig 4 illustrates the MS Spectra data interpretation window, highlighting the analysis of fragment structures used to generate the score, including the count of matches and mismatches.

This window allows for a comparison to determine if the metabolites exhibit a similar fragmentation pattern compared to the substrate fragmentation. Metabolite fragment ions may either share the same m/z as a parent fragment ion (non-shifted ion) or exhibit a defined mass shift (shifted ion).

The MS and MS/MS spectra contain 5 types of fragments:

- **Black peaks:** These peaks lack fragment assignments in the parent, they have no effect on the interpretation of the metabolite under consideration.

- **Red peaks:** Represent matching peaks, and their structural interpretation aligns with the proposed metabolite structure. Clicking on red peaks reveals the assigned structure in the right panel.

- **Cyan peaks:** Indicate mismatching peaks, and their structural interpretation contradicts the proposed metabolite structure.

- **Coral peaks:** Correspond to metabolite matching peaks with structural information consistent with the proposed metabolite structure. However, they lack a substrate fragment match, resulting from manual editing or MassMetaSite if metabolite fragmentation is selected in the settings.

- **Light green peaks:** Denote metabolite mismatching peaks, providing structural information contrary to the proposed structure under study. These peaks lack substrate peak matches and stem from the propagation of a manually edited peak.

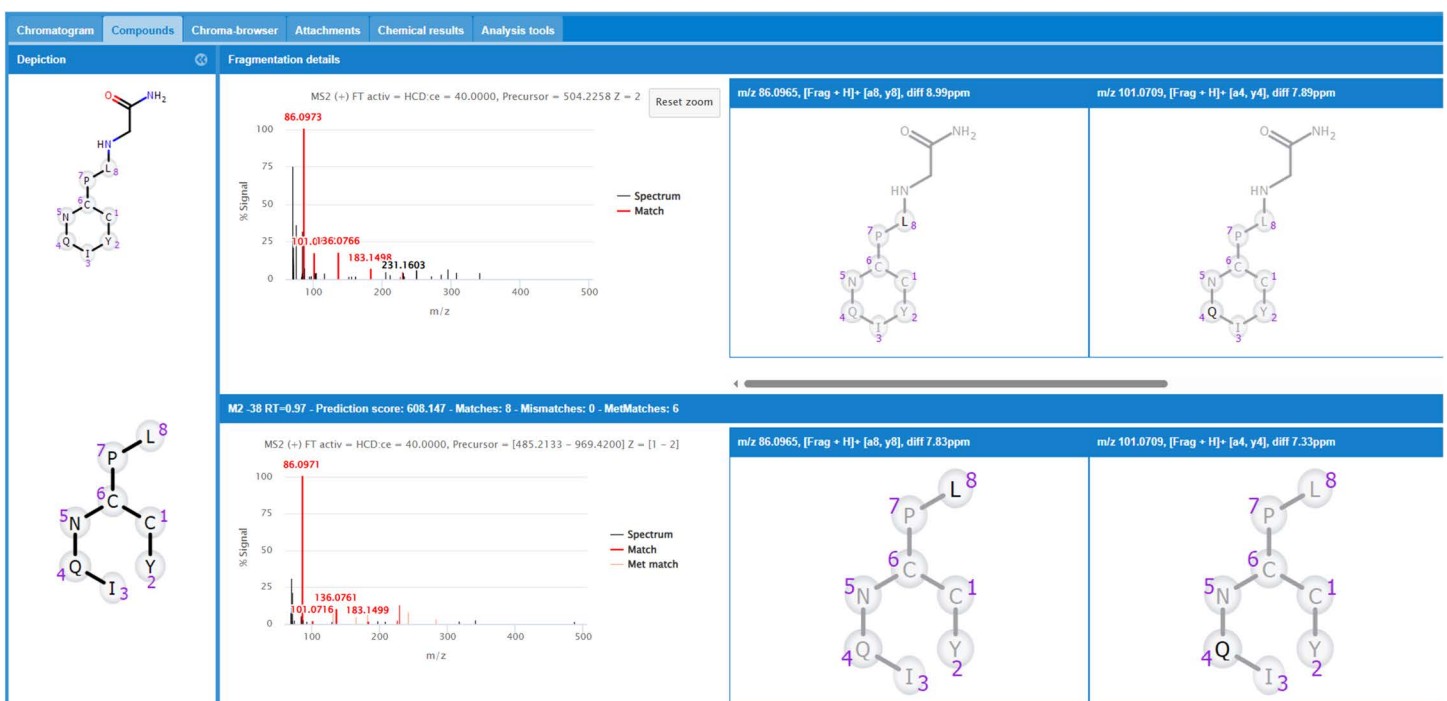

**Fig 4. Fragmentation pattern for the M2-38 metabolite of oxytocin in incubation with chymotrypsin in 120 minutes.** On the left, full scan/ data-dependent MS/MS spectras for oxytocin and M2-38 are presented, while on the right, a subset of fragment structures derived from the selected matched peaks is displayed.

It is essential to consider the isotope pattern and ensure that it aligns with the expected charge state of the metabolite. The charge of the ion significantly influences the spacing between isotopic peaks, and deviations in the observed pattern may serve as indicators of errors in charge assignment or other issues.

Furthermore, the structural assignment of the isotope pattern peaks is checked manually. If the structure assignment of a match or mismatch peak is not the expected one, it can be removed from the analysis and therefore the score will be re-calculated. In addition, black peaks can be examined, and structural information can be added by using the fragment structure editor if it is considered.

**Processing time.** In this study, the data processing time has also been collected, encompassing the duration required for importing data into WebMetabase. Notably, dataset-5 facilitated a comparison with previously reported processing times in the bibliography [15], utilizing the same software with an outdated version (2021). A comparison of the processing time has also been conducted between the different algorithms and settings outlined in the Data Preprocessing section. Since the processing time may vary depending on the peak algorithm employed, as well as the choice of visualization for the compound representation, including expanded, non-expanded, or mixed options.

## Results and discussion

This section presents the experimental results obtained through the application of our approach and algorithms to perform the MetID of the five distinct peptide datasets and an oligonucleotide dataset. All these metabolite structural assignments have been checked manually and considered as reliable because the fragmentation was adequate, isotope pattern was as expected, the m/z small differences between the m/z of observed and theoretical (<10 ppm), and the score was high.

### Monoisotopic mass and most abundant mass

One of the primary objectives of this study is to conduct a comprehensive comparison between the two algorithms, MiM and MaM. To achieve this goal, datasets 1, 2, 3, 4 and 6 as previously outlined, have undergone processing with both algorithm configurations. Table 6 presents the number of identified metabolites corresponding to each dataset, based on the employed algorithm.

Notable differences between MiM and MaM algorithms are observed in compounds such as calcitonin from dataset-1 or taspoglutide from dataset-2. These variations are attributed to the larger peptide structures of these compounds. As molecular size increases, the relative intensity of the MiM tends to decrease. In such cases, the use of the MaM algorithm provides a more precise MetID in larger peptides.

The analysis of dataset-1 resulted in the identification of 150 metabolites through the MiM algorithm, while 161 metabolites were identified using the MaM algorithm. Calcitonin, a cyclic peptide, is one of the largest peptides of this dataset (3429.71 Da), yielding the identification of the same 6 metabolites with both settings, M1-2178, M2-2309, M3-1981, M4-1852, M5-499, and M6-1739 with the respectively retention times of 1.86, 1.91, 2.45, 2.47, 2.52, and 2.99 minutes. However, there is a noticeable difference between them in the score values Table 7. A higher score indicates a better match between the theoretical product ion m/z value and the observed m/z value in the MS/MS spectrum and therefore a more confident structure prediction. This scoring system helps in distinguishing reliable matches from potential false positives.

The dataset-2, consisting of GLP-1 and four synthetic analogues, comprises linear peptides with a molecular weight exceeding 3000 Da, thereby accentuating the significant differences when utilizing MaM or MiM algorithms. This contrast is evident in the case of taspoglutide, as illustrated below.

Taspoglutide (3338.71 Da) incubated with DPP-4 has yielded 15 metabolites peaks with the MaM settings (M1-2175, M2-2163, M3-1966, M4-2223, M5-1925, M6-1895, M7-2222, M8-2154, M9-2255, M10-2094, M11-2147, M12-1977, M13-1396, M14-1146 and M15-407) with a retention time of 2.69, 3.54, 3.74, 3.88, 3.96, 4.00, 4.26, 4.47, 4.48, 4.54, 4.54, 4.92, 5.27, 5.90, and 6.80 respectively. In contrast, using MiM settings, 14 metabolites have been identified, the same as

**Table 6. Number of identified metabolites for each dataset, considering the algorithm, incubation conditions, and acquisition mode (in case of dataset-3).**

| DATASET-1 | INCUBATION CONDITIONS | | | |
|---|---|---|---|---|
| | **Trypsin** | **Chymotrypsin** | **Pancreatic Elastase** | **Pepsin** |
| MiM | 34 | 42 | 39 | 35 |
| MaM | 36 | 45 | 43 | 37 |
| DATASET-2 | INCUBATION CONDITIONS | | | |
| | **DPP-4** | | **NEP** | |
| MiM | 26 | | 4 | |
| MaM | 27 | | 4 | |
| DATASET-3 | ACQUISITION MODE | | | |
| | **DDA** | | **DIA** | |
| MiM | 50 | | 111 | |
| MaM | 50 | | 111 | |
| DATASET-4 | STRUCTURE VISUALIZATION | | | |
| | **NON-EXPANDED** | | | |
| MiM | 7 | | | |
| MaM | 11 | | | |
| DATASET-5 | INCUBATION CONDITIONS | | | |
| | **Trypsin** | **MMP12** | **Neutrophil Elastase** | **CatG** |
| MaM | 31 | 60 | 70 | 77 |
| DATASET-6 | INCUBATION CONDITIONS | | | |
| | **IDE** | | | |
| MiM | 8 | | | |
| MaM | 12 | | | |

**Table 7. Retention times of the identified Calcitonin metabolites along with their corresponding values for score, matches, mismatches, and metmatches obtained using both algorithms.**

| $R_T$ (minutes) | Most abundant mass | | | | Monoisotopic mass | | | |
|---|---|---|---|---|---|---|---|---|
| | **Score** | **Matches** | **Mismatches** | **MetMatches** | **Score** | **Matches** | **Mismatches** | **MetMatches** |
| 1.86 | 445.4 | 2 | 1 | 13 | 126.1 | 1 | 0 | 0 |
| 1.91 | 928.3 | 10 | 0 | 17 | 523.4 | 6 | 0 | 0 |
| 2.45 | 914.2 | 12 | 2 | 28 | 175.5 | 2 | 0 | 1 |
| 2.47 | 1062.3 | 13 | 1 | 40 | 197.1 | 0 | 0 | 0 |
| 2.52 | 1864.0 | 17 | 1 | 19 | 454.9 | 6 | 0 | 0 |
| 2.99 | 1177.4 | 23 | 3 | 30 | 234.5 | 2 | 0 | 0 |

with MaM, but missing M6-1895 (at a retention time of 4.00). Eight of the metabolites correspond to first-generation products (from a single reaction) and are indicated by the green color of the peak, as shown in Fig 5. The other seven brown colored metabolites are indicative of multiple enzymatic reactions. A score is calculated and reported for each metabolite. It can be highlighted that the increased number of matches in the MaM analysis contributes to higher maximum score values. This increase in score values convert a greater level of confidence in the results obtained. As for example, with MaM the metabolite M4-2223 the score is 1302.1 with 28 matching fragments, while with MiM the same metabolite results in a score of 807.1 with 17 matching fragments. Other results are shown in supporting information.

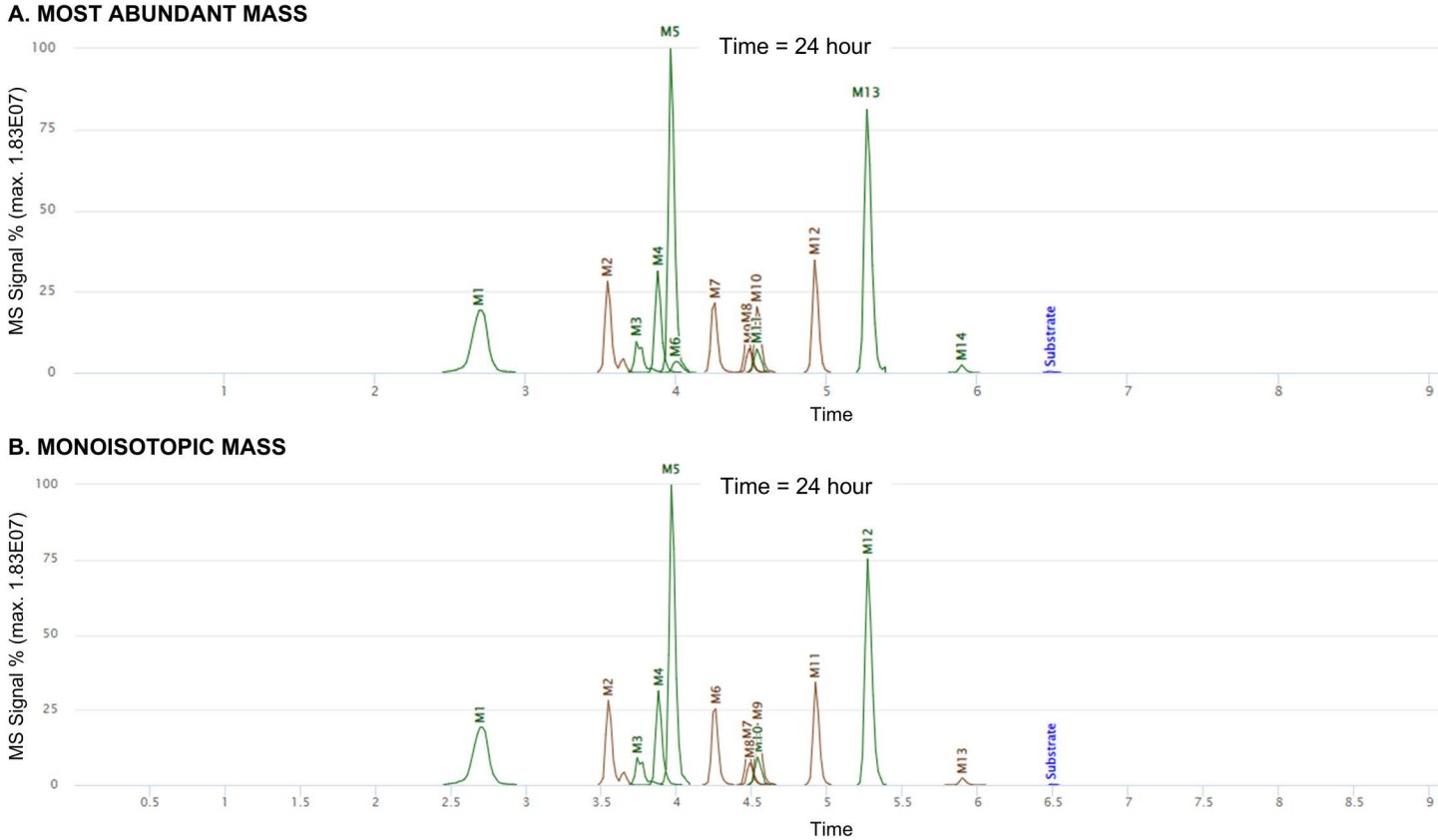

**Fig 5. Extracted ion chromatograms of Taspoglutide after 24 hours of incubation with DPP-4, using both algorithms.** (Blue peak: represents the parent peptide compound, green peaks: first generation of metabolites, and brown peaks: second generation or higher).

The peptide GLP-1 (3297.68 Da) exhibits a brief half-life, primarily attributed to its swift degradation by proteases DPP-4 and NEP. MetID of GLP-1, incubated with DPP-4, revealed the presence of three metabolites: M1-137, M2-394, and M3-208, with respective retention times of 6.53, 6.58, and 6.66 minutes. Notably, M3-208 exhibits the common cleavage site reported in bibliography [24] and attributed to DPP-4, occurring between Ala (8) and Glu (9). A discernible distinction between the two algorithms lies in the appearance of false positives, as shown in Fig 6, with a notable increase observed when employing the MiM settings.

Semaglutide, a GLP-1 analogue, underwent data collection using the HDMS$^E$ acquisition mode on a Waters® QToF instrument. Structural assignments for two degradation products with both algorithms MiM and MaM, namely M1-3446 and M2-3418, have been achieved with high mass accuracy, featuring retention times of 2.77 and 3.07 minutes, respectively (Fig 7). Consistent with prior bibliography, these metabolites arise from three distinct metabolic modifications, specifically induced by amide hydrolysis and sequential beta-oxidation in the fatty acid part [25].

Dataset-3 (comprising somatostatin and seven synthetic analogs incubated with human serum) allows the analysis with different acquisition modes in order to illustrate that the workflow for MetID employing data coming from distinct structural mass spectrometry techniques as DIA and DDA.DDA data was collected with Thermo Scientific Q-Exactive Hybrid Quadrupole-Orbitrap Mass Spectrometer (Q-Exactive) instrument employing full scan mode and DIA HDMS$^E$ data were acquired using a Vion IMS QTof Mass Spectrometer. Both data was processed through Mass-MetaSite, and subsequently uploaded to WebMetabase for visualization via the Mass-MetaSite Batch Processor.

## A. MOST ABUNDANT MASS

| Name | RT ↑ | m/z | Max score | Mass shift | m/z diff [ppm] | m/z diff [mDa] | MiM | Z | MS Area | ion formula | Matches | Mismatches | MetMatches |
|---|---|---|---|---|---|---|---|---|---|---|---|---|---|
| Hidden metabolites | | | | | | | | | | | | | |
| M4 -208 RT=7.05 | 7.04→7.06 | 787.6500→787.6501 | 731.7 | -208.0960 | 1.15→1.35 | 0.91→1.06 | 3,145.5720 | 4 | 2.41E06→5.01E06 | [C$_{142}$H$_{216}$N$_{36}$O$_{45}$+... | 6 | 0 | 1 |
| M5 -2326 RT=11.13 | 11.09→11.13 | 258.1150→258.1152 | 383.1 | -2,325.2382 | -1.86→-1.08 | -0.48→-0.28 | 1,028.4298 | 4 | 2.94E05→3.96E05 | [C$_{42}$H$_{64}$N$_{10}$O$_{20}$+H... | 3 | 0 | 1 |
| M6 -2118 RT=11.35 | 11.34→11.36 | 413.1846→413.1850 | 183.2 | -2,117.1422 | -5.97→-4.92 | -2.47→-2.03 | 1,236.5259 | 3 | 2.89E05→8.71E05 | [C$_{51}$H$_{76}$N$_{14}$O$_{22}$+H... | 3 | 0 | 0 |
| M7 -2326 RT=11.42 | 11.42→11.51 | 258.1153→258.1154 | 255.5 | -2,325.2382 | -2.63→-2.25 | -0.68→-0.58 | 1,028.4298 | 4 | 3.48E05→4.43E05 | [C$_{42}$H$_{64}$N$_{10}$O$_{20}$+H... | 3 | 0 | 1 |
| M8 -2326 RT=11.75 | 11.71→11.78 | 258.1152→258.1153 | 409.1 | -2,325.2382 | -2.34→-1.86 | -0.60→-0.48 | 1,028.4298 | 4 | 4.05E05→4.83E05 | [C$_{42}$H$_{64}$N$_{10}$O$_{20}$+H... | 4 | 0 | 1 |
| M9 -2028 RT=11.93 | 11.93→11.99 | 443.2046→443.2049 | 159.1 | -2,027.0741 | 0.83→1.56 | 0.37→0.69 | 1,326.5939 | 3 | 4.62E05→6.15E05 | [C$_{55}$H$_{86}$N$_{14}$O$_{24}$+H... | 3 | 0 | 1 |
| M10 -2326 RT=11.93 | 11.93 | 258.1153 | 168.6 | -2,325.2382 | -2.06 | -0.53 | 1,028.4298 | 4 | 4.44E05 | [C$_{42}$H$_{64}$N$_{10}$O$_{20}$+H... | 3 | 0 | 1 |

## B. MONOISOTOPIC MASS

| Name ↑ | RT | m/z | Max score | Mass shift | m/z diff [ppm] | m/z diff [mDa] | MiM | Z | MS Area | ion formula | Matches | Mismatches | MetMatches |
|---|---|---|---|---|---|---|---|---|---|---|---|---|---|
| Hidden metabolites | | | | | | | | | | | | | |
| M1 -2306 RT=1.81 | 1.81→1.89 | 524.7988→524.7991 | 275.4 | -2,306.0927 | -7.94→-7.36 | -4.17→-3.87 | 1,047.5753 | 2 | 1.81E05→2.75E05 | [C$_{62}$H$_{77}$N$_{11}$O$_{42}$+H... | 3 | 0 | 1 |
| M10 -2105 RT=6.94 | 6.93→6.94 | 417.2090→417.2093 | 475.1 | -2,105.0542 | 6.28→6.96 | 2.62→2.91 | 1,248.6139 | 3 | 5.93E04→3.83E05 | [C$_{58}$H$_{84}$N$_{14}$O$_{42}$+H... | 3 | 0 | 1 |
| M11 -208 RT=7.05 | 7.04→7.05 | 787.3990→787.3991 | 787.9 | -208.0960 | 1.53→1.61 | 1.21→1.27 | 3,145.5720 | 4 | 1.38E06→2.89E06 | [C$_{142}$H$_{216}$N$_{36}$O$_{45}$+... | 7 | 0 | 9 |
| M12 -2121 RT=7.14 | 7.13→7.15 | 617.3374→617.3377 | 114.0 | -2,121.0127 | -4.47→-3.96 | -2.76→-2.44 | 1,232.6554 | 2 | 1.76E05→2.10E05 | [C$_{59}$H$_{88}$N$_{14}$O$_{15}$+H... | 3 | 0 | 0 |
| M13 -2325 RT=11.13 | 11.09→11.13 | 258.1150→258.1152 | 318.0 | -2,325.2382 | -1.80→-1.03 | -0.46→-0.26 | 1,028.4298 | 4 | 2.94E05→3.96E05 | [C$_{42}$H$_{64}$N$_{10}$O$_{20}$+H... | 3 | 0 | 1 |
| M14 -2325 RT=11.35 | 11.35→11.45 | 258.1153→258.1154 | 167.1 | -2,325.2382 | -2.58→-2.00 | -0.66→-0.52 | 1,028.4298 | 4 | 1.73E05→2.90E05 | [C$_{42}$H$_{64}$N$_{10}$O$_{20}$+H... | 3 | 0 | 2 |
| M15 -2117 RT=11.36 | 11.34→11.36 | 413.1846→413.1850 | 118.1 | -2,117.1422 | -5.90→-4.85 | -2.44→-2.01 | 1,236.5259 | 3 | 2.89E05→8.71E05 | [C$_{51}$H$_{76}$N$_{14}$O$_{22}$+H... | 3 | 0 | 0 |
| M16 -2325 RT=11.42 | 11.42→11.51 | 258.1153→258.1154 | 190.4 | -2,325.2382 | -2.58→-2.19 | -0.66→-0.56 | 1,028.4298 | 4 | 3.48E05→4.43E05 | [C$_{42}$H$_{64}$N$_{10}$O$_{20}$+H... | 3 | 0 | 1 |
| M17 -2325 RT=11.75 | 11.71→11.78 | 258.1152→258.1153 | 244.0 | -2,325.2382 | -2.28→-1.80 | -0.59→-0.46 | 1,028.4298 | 4 | 4.05E05→4.83E05 | [C$_{42}$H$_{64}$N$_{10}$O$_{20}$+H... | 3 | 0 | 2 |
| M18 -2027 RT=11.93 | 11.93→11.99 | 443.2046→443.2049 | 94.0 | -2,027.0741 | 0.80→1.53 | 0.36→0.68 | 1,326.5939 | 3 | 4.62E05→6.15E05 | [C$_{55}$H$_{86}$N$_{14}$O$_{24}$+H... | 3 | 0 | 1 |
| M19 -2325 RT=11.93 | 11.93 | 258.1153 | 102.7 | -2,325.2382 | -2.13 | -0.55 | 1,028.4298 | 4 | 4.06E05 | [C$_{42}$H$_{64}$N$_{10}$O$_{20}$+H... | 3 | 0 | 1 |
| M2 -2150 RT=2.84 | 2.78→3.01 | 301.8879→301.8880 | 99.6 | -2,150.1385 | 5.51→5.84 | 1.66→1.76 | 1,203.5296 | 4 | 1.47E05→1.68E05 | [C$_{63}$H$_{77}$N$_{11}$O$_{21}$+H... | 3 | 0 | 6 |
| M3 -2265 RT=5.24 | 5.22→5.26 | 273.1669→273.1670 | 131.8 | -2,265.0186 | 9.69→10.06 | 2.65→2.75 | 1,088.6495 | 4 | 6.21E05→3.21E06 | [C$_{65}$H$_{84}$N$_{14}$O$_{16}$+H... | 3 | 0 | 0 |
| M4 -2150 RT=5.40 | 5.28→5.40 | 301.8881→301.8882 | 88.4 | -2,150.1385 | 4.76→5.18 | 1.44→1.56 | 1,203.5296 | 4 | 1.31E05→1.92E05 | [C$_{63}$H$_{77}$N$_{11}$O$_{21}$+H... | 3 | 0 | 4 |
| M8 -265 RT=6.65 | 6.65→6.67 | 773.1417→773.1429 | 273.9 | -265.1175 | 2.61→4.16 | 2.02→3.22 | 3,088.5506 | 4 | 2.60E04→7.23E05 | [C$_{140}$H$_{213}$N$_{35}$O$_{44}$+... | 6 | 0 | 2 |

**Fig 6. False positives of the GLP-1 compound using both the MaM and MiM algorithms.** It is noteworthy that the number obtained with the MiM algorithm is significantly higher.

DDA and DIA data underwent processing with both algorithms (MiM and MaM). The results obtained show no distinctions. The identified metabolites, score values, and various parameters such as the numbers of matches, mismatches, and metmatches remain consistent across both algorithms. Considering the minimal chemical or monomer modifications within the peptide structure of these compounds, no substantial shift in molecular size was observed in this dataset.

The analysis of this dataset collected with DDA led to the identification of 17 metabolites for each of the algorithms. All the metabolites identified have been produced from amide hydrolysis reaction. The principal metabolite formations observed include the generation of -Ala (−71 Da) and -AlaGly (−128 Da) from the linear segment of the structure (Fig 8). The incorporation of D-Trp at the eighth position showed an improved stability over the parent compound somatostatin, due to the differences in the appearance of metabolism as synthetic analogs avoid the ring opening observed between D-Trp (8) and Lys (9). This observation aligns with findings from previous bibliography, which highlighted that the introduction of Msa residues, coupled with the presence of D-Trp8, contributes to the augmentation of aromatic side-chains interactions in Somatostatin, providing a greater stability [13].

Similarly, for DIA data, the identification of key metabolites, specifically -Ala and -AlaGly, is consistent. As previously documented in bibliography [26], the analog labeled as 95 demonstrates superior stability, characterized by delayed and reduced metabolic transformations compared to other analogs. This stability is further elucidated in Fig 9, which delineates the time/response profiles of the substrate, illustrating the gradual disappearance of the peptide.

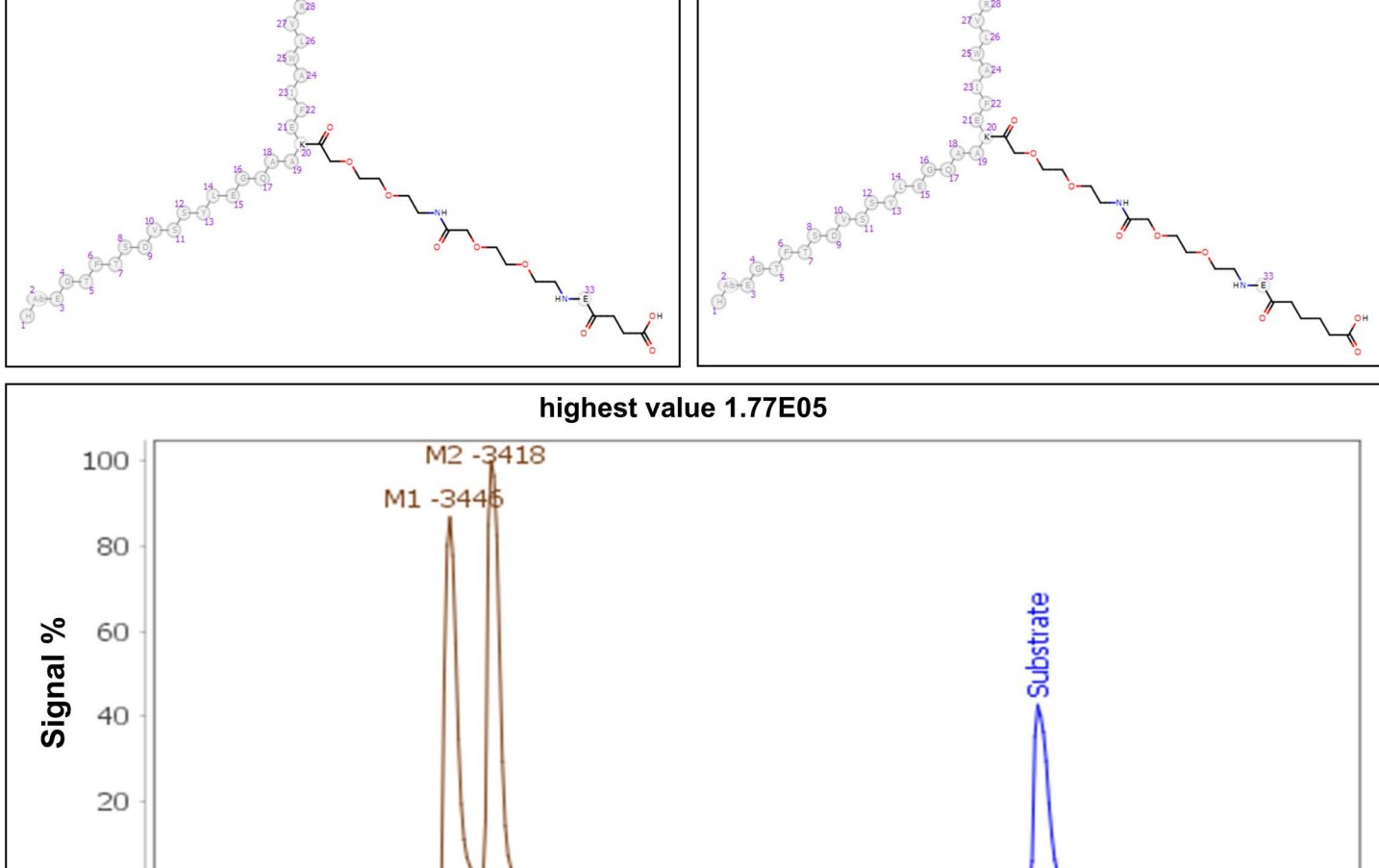

**Fig 7. Metabolites identified and extracted ion chromatogram of Semaglutide using MiM algorithm.**

Dataset-5 contains 16 linear and 12 cyclic peptides, incubated with cathepsin G, neutrophil elastase, trypsin and MMP-12. The data was collected using LC-HRMS, with analysis performed on a Synapt G2® high-definition quadrupole time-of-flight mass spectrometer (Waters®), operating in positive electrospray ionization modeThe data processing time, employing the settings outlined in the referenced research [13] study and utilizing the non-expanded structure visualization, has undergone a substantial reduction. As an illustration, the compound salmon calcitonin, which conventionally needed two hours for processing, now, requires only 25 minutes with the implementation of the new methodology.

As an illustrative example of this dataset, the following compound and its analogs will be described, while the MetID for the other compounds can be found in the Supplementary Information. Specifically, the dataset includes somatostatin, and analogs that have been synthesized over the past few decades introducing modifications such as exchange and deletion of amino acids, ring size reduction, or disulfide bridge modification, among others. [13] These analogs, namely octreotide,

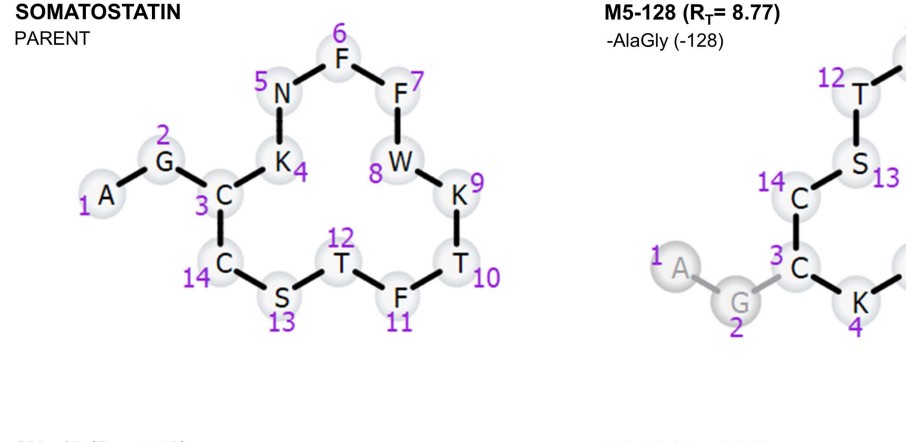

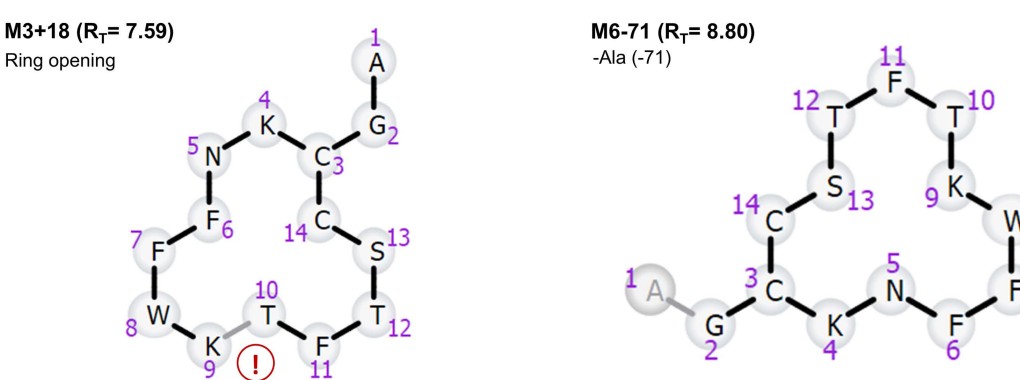

**Fig 8. Somatostatin (Parent compound) and major metabolites identified using both algorithms.** Metabolites M5-128 and M6-71 indicate cleavages from the tail portion of somatostatin. Additionally, M3 + 18 represents a ring-opening product occurring between DTrp (8) and Lys (9).

lanreotide, and vapreotide, are octapeptides characterized by a shorter and consequently less flexible ring structure compared to somatostatin.

Previous bibliography reports that the ring opening from somatostatin and its analogs was only observed in the case of somatostatin, as also observed in this study [13]. Despite somatostatin being rapidly degraded by proteases, its analogs exhibit stability, as illustrated in Fig 10, which presents extracted ion chromatograms after 60 minutes of incubation with neutrophil elastase. The processing time for these compounds was 15 minutes.

Fig 11 presents a detailed MetID of somatostatin incubated with neutrophil elastase. The analysis identified the same metabolites as reported in the previously bibliography [13]: M1-1371, M2-1204, M3-230, M4 + 18, M5-909, M6 + 18, and M7-661, with respective retention times (RT) of 0.73, 1.60, 1.60, 1.71, 1.93, 1.93, and 2.21.

Dataset-6 contains data of human insulin (5808 Da), a cyclic peptide with three disulfide bridges, after the incubation with IDE at 2 minutes. Computing using the MaM algorithm led to the identification of 12 metabolites, designated as M1-2965, M2-3315, M3-3145, M4-2973, M5-2902, M6-3452, M7-3151, M8-3032, M9-2961, M10-3289, M11-2869, and M12-2798, with respective retention times of 2.06, 7.78, 8.08, 9.14, 9.65, 9.80, 10.38, 11.17, 11.43, 11.69, and 12.39 minutes (Fig 11). These metabolites have been previously documented in the bibliography and are generated through two cleavages, one within Chain A and the other within Chain B. Notably, four of them have been reported previously as major IDE-degraded insulin fragments (Fig 12) [16]. The formation of these metabolites results from cleavage occurring either within the A chain, specifically at positions A13-14 or A14-15, and in the middle of the B chain, either at positions B9-10 or B14-15.

## A. SOMATOSTATIN

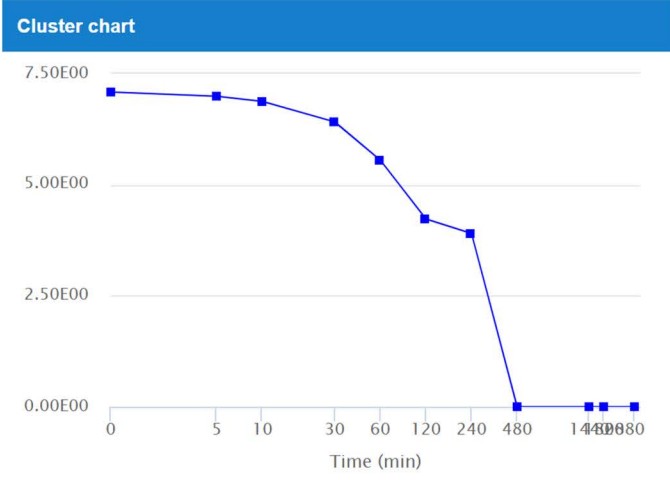

## C. ANALOGUE 31

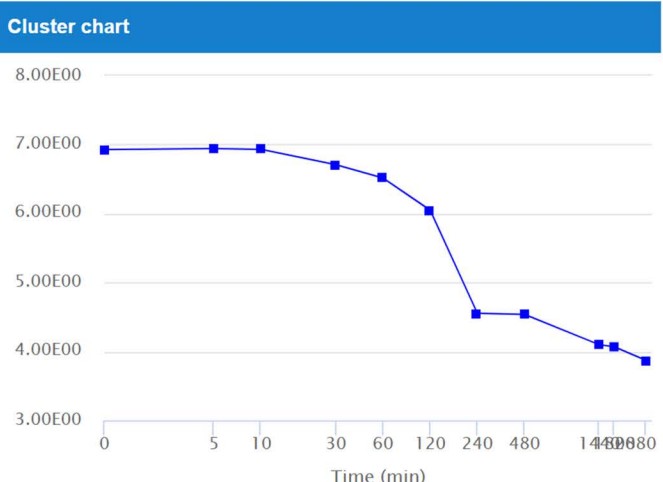

## B. ANALOGUE 65

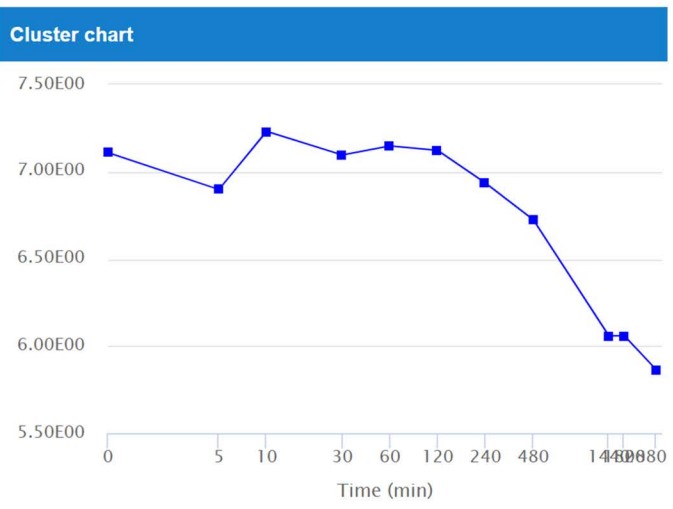

## D. ANALOGUE 95

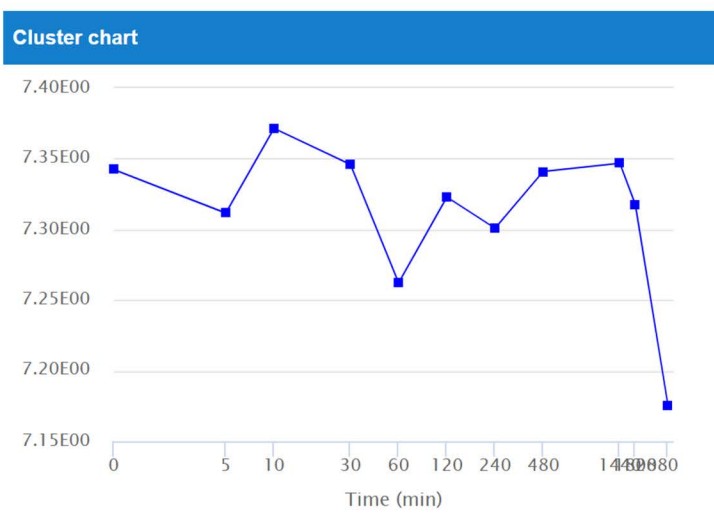

**Fig 9. Substrate profiles employing ln-ln scaling for somatostatin, Analogue 31, 65, and 95.**

In contrast, MiM identified 8 metabolites, M1-3306, M2-2971, M3-3450, M4-3150, M5-2959, M6-3287, M7-2867, and M8-2618, with respective retention times of 7.74, 9.14, 9.65, 9.78, 11.15, 11.43, 11.67, and 15.65 (Fig 13). Notably, two of the major previously bibliography-reported products are absent [16]. Moreover, consistent with previous observations, there is a significant difference in score values between the two algorithms, with MaM. scores consistently higher due to the higher number of matches and no presence of mismatches Table 8.

### Structure visualization – atoms/bonds vs monomer

The analysis of biotransformation products for therapeutic oligonucleotides using LC-HRMS presents a significant challenge, primarily attributed to the high molecular weight of these compounds. Given that these oligonucleotides consist of multiple monomers susceptible to metabolic reactions, constructing a virtual set containing all potential metabolites becomes a resource-intensive task in terms of time and computational requirements. Furthermore, the extensive number

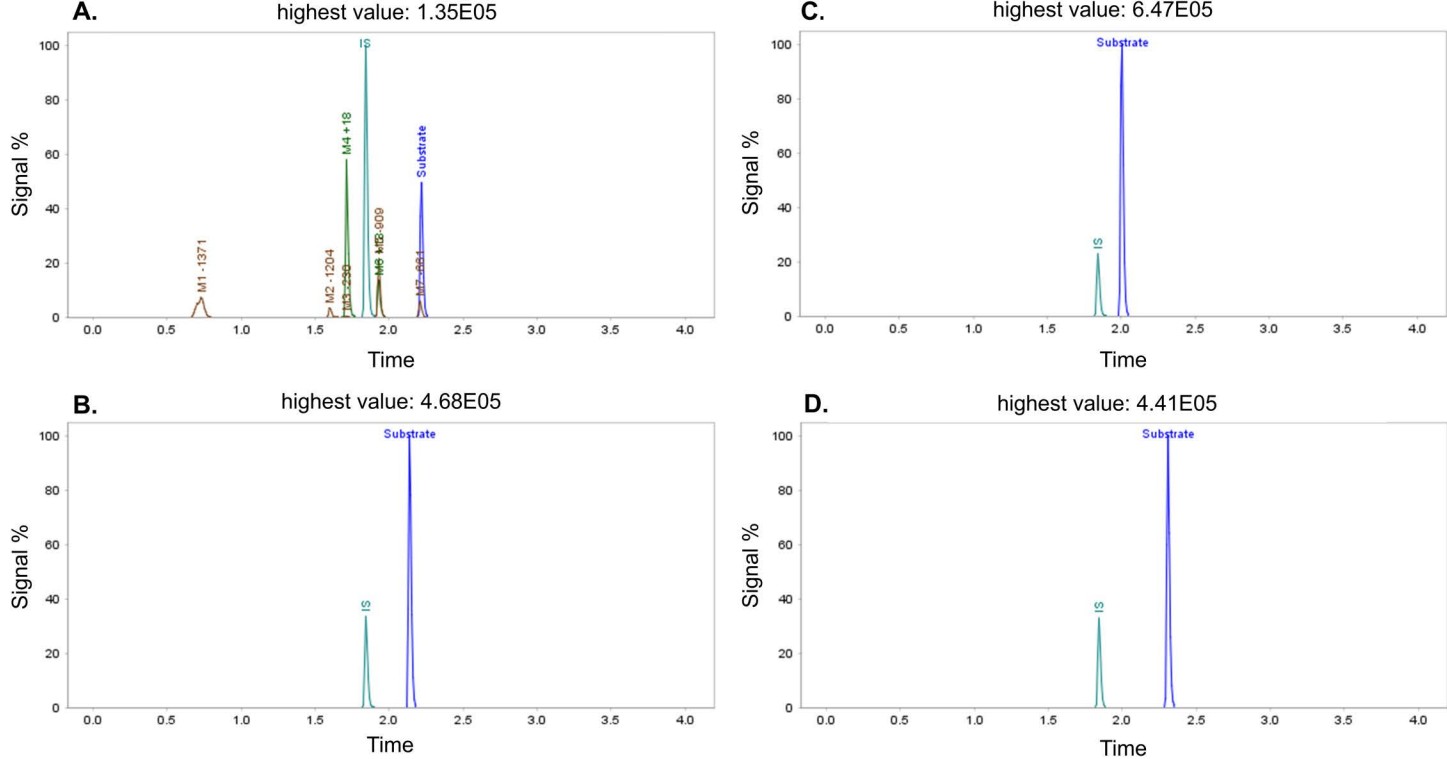

**Fig 10. Extracted ion chromatograms using MaM algorithms.** A) somatostatin, B) lanreotide, C) octreotide, and D) vapeotride after 60 minutes of incubation with neutrophil elastase.

of cleavable bonds amplifies the complexity of the fragmentation analysis, demanding additional time and computing resources. This study shows the fragmentation algorithm that allows the analysis at monomer levels (non-expanded) and the other at the atom/bond levels (expanded).

In this section, three experiments involving the incubation of ASOs in Human Liver at various timepoints are presented, comprising two sets incubated with distinct oligonucleotide strains (dataset-4). The data was acquired in a DDA mode in a Thermo Q-Exactive® spectrometer.

A total of 11 metabolites have been identified in both experiments (expanded and non-expanded) using the MaM algorithm, M1-5473, M2-5473, M3-3282, M4-2567, M5-930, M6-617, M7-616, M8-313, M9-312, M10-304, M11-304, with respective retention times of 7.17, 8.62, 14.42, 16.36, 17.11, 17.59, 17.84, 17.90, 17.92, 17.95, and 18.26 (Fig 14). The identified structures of the metabolites can be attributed to specific biotransformation reactions, including o-dealkylation, phosphoester hydrolysis, aromatic deamination, and nucleobase loss.

In contrast, using the MiM algorithm, a total of 7 metabolites have been identified (using non-expanded visualization), M1-5470, M2-5470, M3-2566, M4-617, M5-313, M6-313, and M7-304, with respective retention times of 7.17, 8.62, 16.36, 17.55, 17.89, 17.93, and 18.24. Table 9 illustrates the score value differences between the two algorithms.

In addition, a paired-samples *t*-test was performed on the Score values reported in Tables 7–9 (excluding metabolites not found in one of the two approaches), revealing a statistically significant difference between the MiM and MaM algorithms ($p = 0.0002904$).

In Fig 15, the two distinct structure visualizations are presented for the same identified metabolite, showcasing a nucleobase loss from the parent compound and two phosphoester hydrolyses. The depiction at the bond level provides

| Peak Name | Structure Proposal | m/z | ppm | Area % | Conditions |
|---|---|---|---|---|---|
| M1 -1371<br>RT=0.73 | | 267.1337 | 3.7 | 0.91 | Time (min)=23min |
| | | | -2.5 | 0.07 | Time (min)=12min |
| | | | 3.1 | 9.78 | Time (min)=60min |
| | | | 1.7 | 0.40 | Time (min)=17min |
| | | | -2.1 | 0.02 | Time (min)=6min |
| M2 -1204<br>RT=1.60 | | 434.2419 | 7.3 | 1.83 | Time (min)=60min |
| | | | -12.1 | 0.06 | Time (min)=6min |
| | | | -11.2 | 0.12 | Time (min)=12min |
| | | | -6.3 | 0.35 | Time (min)=23min |
| | | | -2.4 | 0.24 | Time (min)=17min |
| | | | -3.3 | 0.06 | Time (min)=0min |
| M3 -230<br>RT=1.60 | | 469.8776 | 4.5 | 1.71 | Time (min)=23min |
| | | | 4.8 | 2.73 | Time (min)=17min |
| | | | -0.6 | 0.21 | Time (min)=6min |
| | | | -0.8 | 0.05 | Time (min)=12min |
| | | | -2.1 | 0.01 | Time (min)=0min |
| | | | -5.0 | 0.20 | Time (min)=60min |
| M4 +18<br>RT=1.71 | | 414.6798 | 47.8 | 28.00 | Time (min)=60min |
| | | | 8.8 | 4.95 | Time (min)=12min |
| | | | 17.8 | 10.02 | Time (min)=23min |
| | | | 15.9 | 6.66 | Time (min)=17min |
| | | | 36.1 | 2.41 | Time (min)=6min |
| | | | 7.2 | 0.24 | Time (min)=0min |
| M5 -909<br>RT=1.93 | | 364.6903 | 1.2 | 11.03 | Time (min)=60min |
| | | | 3.8 | 0.30 | Time (min)=12min |
| | | | 3.1 | 1.31 | Time (min)=23min |
| | | | 5.8 | 0.58 | Time (min)=17min |
| | | | 8.5 | 0.06 | Time (min)=6min |
| M6 +18<br>RT=1.93 | | 414.6851 | 4.9 | 14.38 | Time (min)=60min |
| | | | 24.3 | 7.05 | Time (min)=12min |
| | | | 7.4 | 9.36 | Time (min)=17min |
| | | | 0.2 | 0.48 | Time (min)=0min |
| | | | 23.6 | 4.30 | Time (min)=6min |
| | | | -3.1 | 0.08 | Time (min)=23min |
| M7 -661<br>RT=2.21 | | 488.7493 | 5.6 | 1.75 | Time (min)=23min |
| | | | -2.8 | 3.28 | Time (min)=60min |
| | | | 2.0 | 0.94 | Time (min)=17min |
| | | | 7.4 | 0.14 | Time (min)=6min |
| | | | -4.9 | 0.60 | Time (min)=12min |

**Fig 11. Summarized MetID reports which each retention time (RT) from incubation of somatostatin with neutrophil elastase.**

a clearer understanding of the biotransformation pathways and chemical alterations experienced by the compound. It is noteworthy to consider the processing time, which, in this specific example, is 40 minutes for the non-expanded representation and extends to 70 minutes when three of the monomers are expanded.

This visualization algorithm allows to combine monomer and atom/bond notation, being then easily to see the metabolic changes in the structure. As a result, the need to expand all monomers individually is avoided, alleviating the associated high processing time. The constraint structure alignment between the substrate and the metabolite, maintaining the same orientation, allows for the interpretation of the occurred biotransformations.

## Processing time

The processing time is influenced by the size and molecular weight of the peptide, as shown in Table 10. For peptides with molecular weights between 3000 and 4000 Da, processing times range from 22 to 30 minutes when using the non-expanded visualization mode. In contrast, the expanded mode results in longer processing times, extending from

## A. MOST ABUNDANT MASS

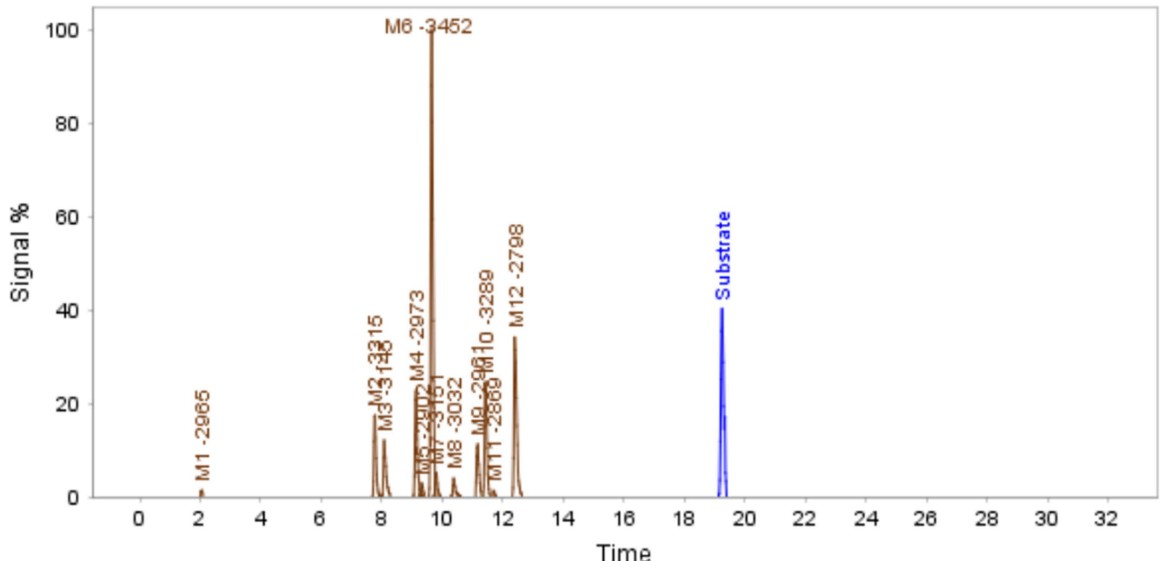

## B. MONOISOTOPIC MASS

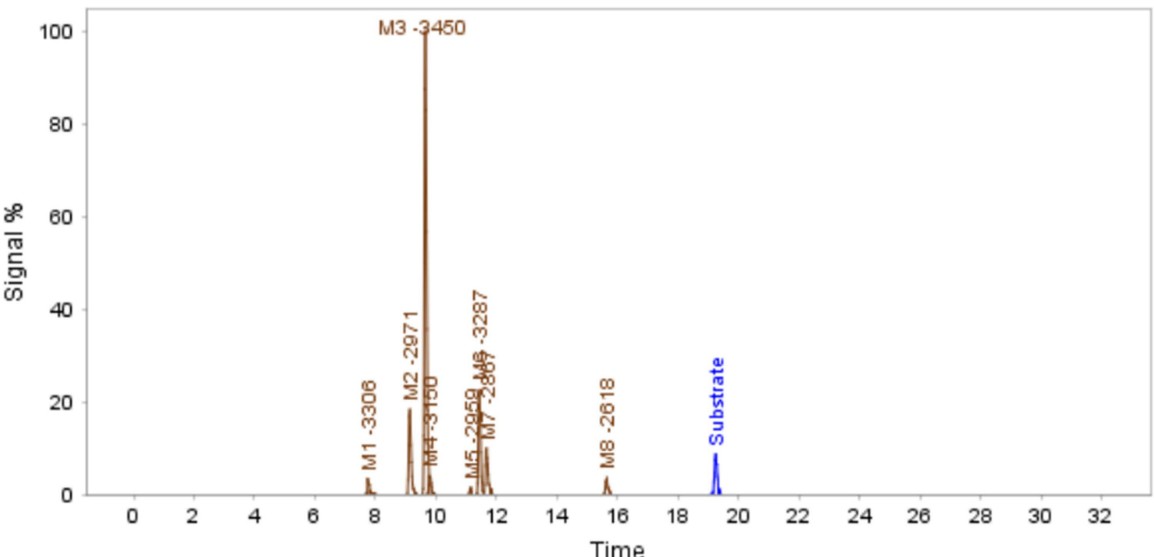

**Fig 12. Extracted ion chromatograms of Insulin after 2 minutes of incubation with IDE.** Blue peak: substrate/parent peptide, green peaks: first generation metabolites, brown peaks: second generation or higher metabolites.

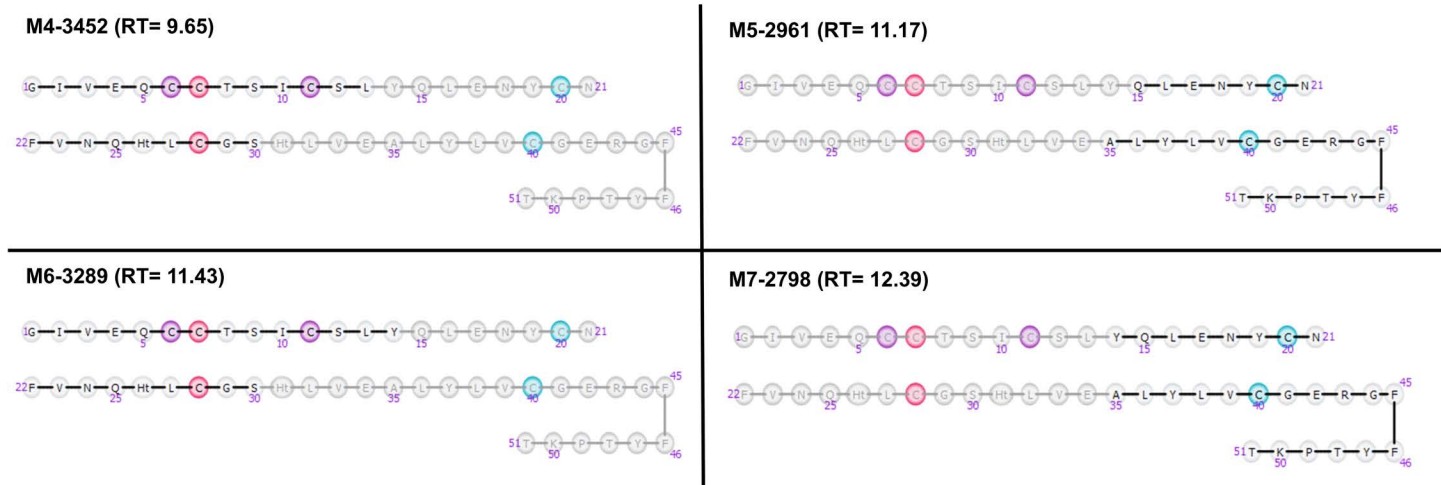

**Fig 13. Four of major products corresponding to Insulin fragments, using MaM algorithm, after incubatution with IDE.** These metabolites, resulting from two distinct cleavages—one within Chain A and the other within Chain B—have been previously identified in the bibliography.

**Table 8. Retention times of the identified Insulin metabolites along with their corresponding values for score, matches, mismatches, and metmatches obtained using both algorithms. NI = Non-identified metabolites.**

| $R_T$ (minutes) | Most abundant mass | | | | Monoisotopic mass | | | |
|---|---|---|---|---|---|---|---|---|
| | Score | Matches | Mismatches | MetMatches | Score | Matches | Mismatches | MetMatches |
| 2.06 | 509.6 | 3 | 0 | 0 | NI | NI | NI | NI |
| 7.78 | 1017.5 | 6 | 0 | 18 | 132.9 | 2 | 0 | 12 |
| 8.08 | 760.3 | 6 | 0 | 5 | NI | NI | NI | NI |
| 9.14 | 816.3 | 6 | 0 | 8 | 253.3 | 4 | 0 | 9 |
| 9.32 | 767.5 | 6 | 0 | 2 | NI | NI | NI | NI |
| 9.65 | 767.1 | 6 | 0 | 9 | 243.6 | 4 | 2 | 10 |
| 9.80 | 874.0 | 6 | 0 | 4 | 278.7 | 4 | 0 | 5 |
| 10.38 | 992.6 | 9 | 0 | 2 | NI | NI | NI | NI |
| 11.17 | 792.4 | 6 | 0 | 4 | 319.0 | 2 | 0 | 3 |
| 11.43 | 737.5 | 6 | 0 | 4 | 223.8 | 4 | 0 | 4 |
| 11.69 | 509.6 | 3 | 0 | 0 | 210.1 | 4 | 1 | 7 |
| 12.39 | 789.8 | 6 | 0 | 8 | NI | NI | NI | NI |
| 15.65 | NI | NI | NI | NI | 138.9 | 2 | 0 | 0 |

2 up to 8 hours. For peptides exceeding 4000 Da, the expanded mode becomes impractical due to excessive memory requirements and long processing times.

This difference is due to the fragmentation method used in each mode: the non-expanded mode operates at the monomer level, limiting fragmentation to predefined ion types (e.g., a, b, c, x, y, z ions in peptide analysis), while the expanded mode simulates fragmentation at the atomic level by disconnecting all chemical bonds. As a result, the expanded approach generates a significantly higher number of theoretical fragments, increasing processing times.

## A. MOST ABUNDANT MASS       highest value 2.32E07

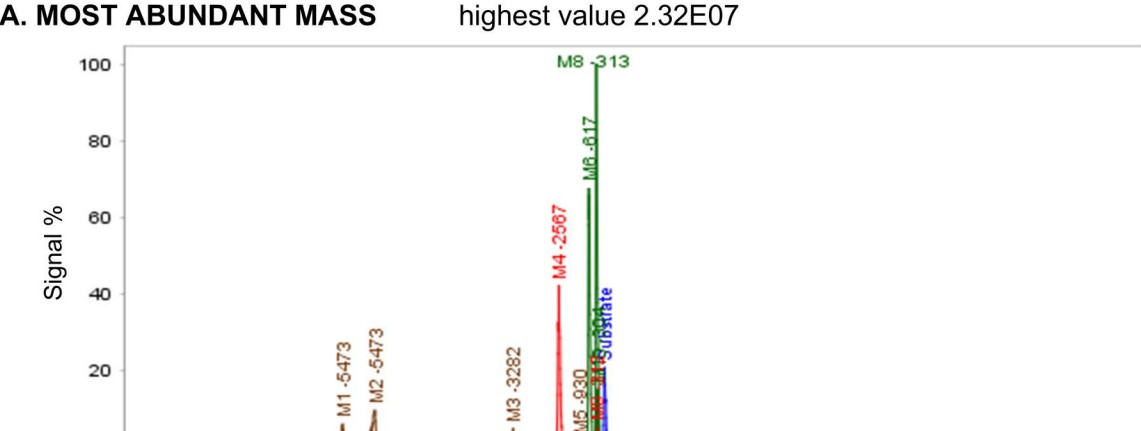

## B. MONOISOTOPIC MASS       highest value 8.51E06

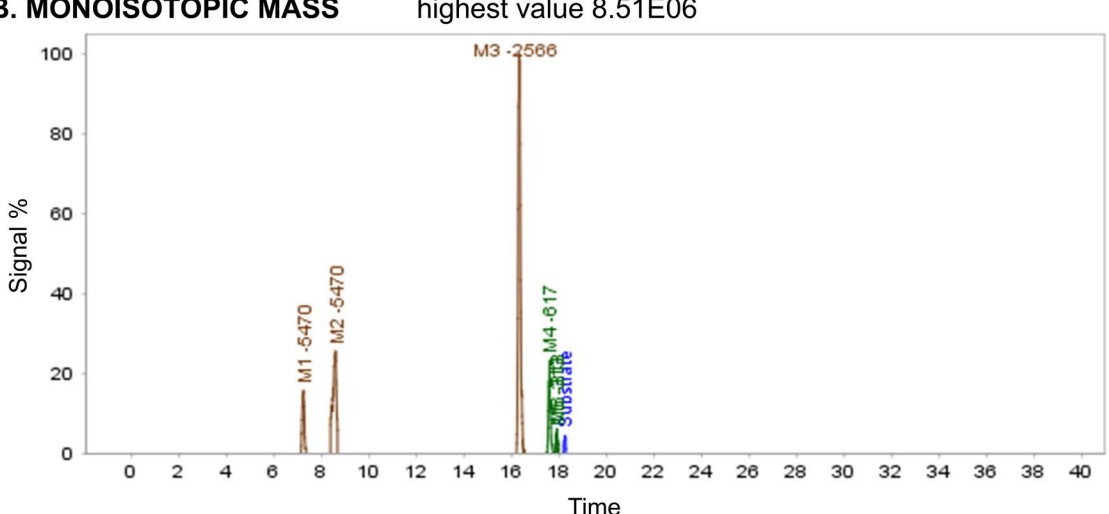

**Fig 14.  Extracted ion chromatograms of ASOs after 72 hours of incubation with the modified strain.**

**Table 9.  Retention times of the identified ASO metabolites along with their corresponding values for score, matches, mismatches, and met-matches obtained using both algorithms. NI = Non-identified metabolites.**

| $R_T$ (minutes) | Most Abundant Mass | | | | Monoisotopic Mass | | | |
|---|---|---|---|---|---|---|---|---|
| | Score | Matches | Mismatches | MetMatches | Score | Matches | Mismatches | MetMatches |
| 7.17 | 2654.9 | 31 | 0 | 2 | 835.9 | 15 | 0 | 0 |
| 8.62 | 2547.5 | 32 | 0 | 5 | 896.8 | 15 | 0 | 0 |
| 14.42 | 2250.2 | 31 | 0 | 11 | NI | NI | NI | NI |
| 16.36 | 4445.7 | 62 | 6 | 33 | 788.3 | 35 | 0 | 3 |
| 17.11 | 4019.8 | 46 | 4 | 19 | NI | NI | NI | NI |
| 17.59 | 5056.3 | 67 | 3 | 18 | 426.1 | 25 | 0 | 2 |
| 17.84 | 1778.9 | 24 | 0 | 0 | NI | 35 | 0 | 5 |
| 17.90 | 6171.2 | 101 | 6 | 19 | 541.9 | NI | NI | NI |
| 17.92 | 4081.9 | 62 | 1 | 4 | 561.3 | 5 | 0 | 3 |
| 17.95 | 8142.5 | 112 | 5 | 38 | NI | NI | NI | NI |
| 18.26 | 5423.3 | 83 | 2 | 21 | 371.5 | 25 | 0 | 1 |

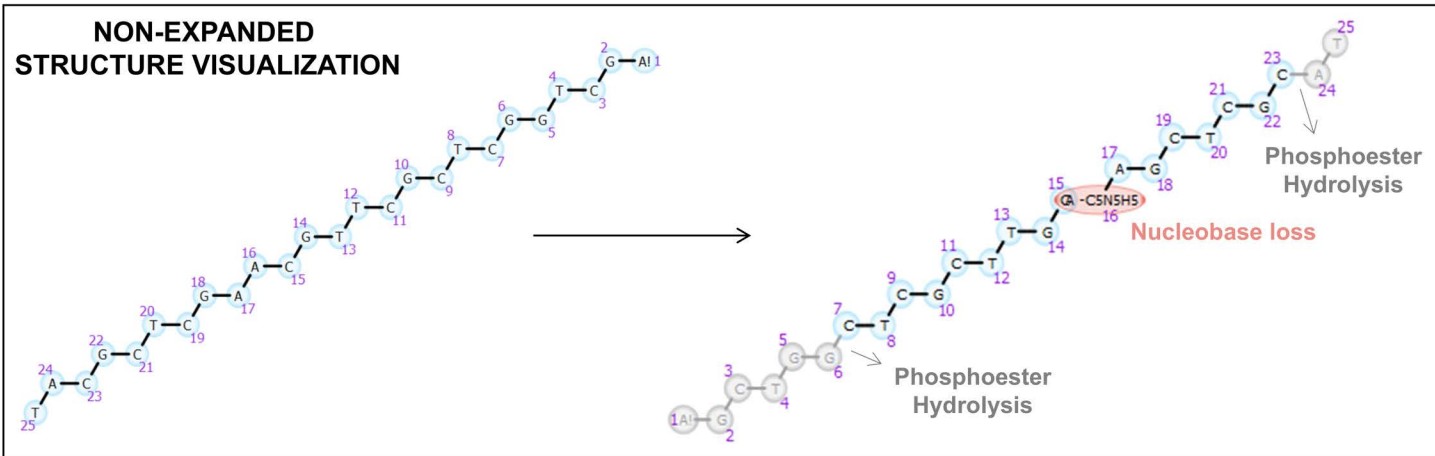

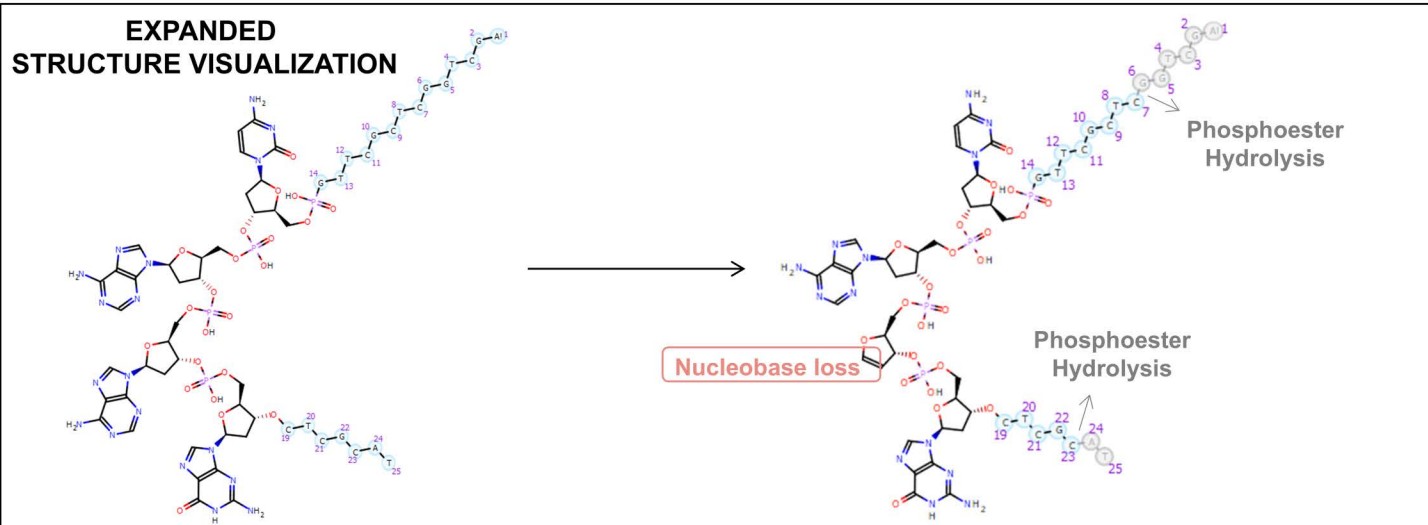

**Fig 15. Illustration of nucleobase loss in both expanded and non-expanded structural representations of ASO.**

**Table 10. Estimated processing times for peptides based on molecular weight and the used visualization approach.**

| Molecular weight (Da) | Number of compounds | All monomers non-expanded (minutes) | All monomers expanded |
|---|---|---|---|
| < 1000 | 5 | 5 - 8 | 25 - 30 minutes |
| 1000 - 1200 | 15 | 8 - 13 | 30 - 40 minutes |
| 1200 - 1500 | 7 | 13 - 18 | 40 - 60 minutes |
| 1500 - 3000 | 12 | 18 - 22 | 60 - 120 minutes |
| 3000 - 4000 | 6 | 22 - 30 | 2–8 hours |
| > 4000 | 4 | 30 - 40 | * |

* Not computable due to high memory requirements and extended processing time.

## Conclusions

A new automated workflow for LC-HRMS data analysis has been described and developed, addressing challenges associated with result visualization and computational time in processing incubated data of macromolecules. This approach has effectively proved the analysis of both linear and cyclic peptides containing natural or unnatural amino acids. A total of 970 metabolites have been identified across different incubation conditions and peak detection algorithms. Furthermore, its applicability extends beyond peptides, as demonstrated by successful processing of oligonucleotide data. The results have shown that the workflow can efficiently manage experimental data within a molecular range spanning 700–7630 Da. Importantly, its effectiveness has been validated across multiple acquisition modes, as data coming from different acquisition modes (DDA and DIA) has been processed.

WebMetabase was employed for the processing and visualization of data derived from six databases using different algorithms in the data preprocessing step.

In larger molecules (>3000 Da), notable differences were observed between the MiM and MaM peak detection algorithms. The MaM approach identified a greater number of metabolites, including several that were missed by MiM but previously reported in the literature, as for example in the case of insulin. In these high-mass compounds, the MaM algorithm produced higher scoring and more numerous matches, indicating increased confidence in structural predictions. Additionally, it demonstrated a lower incidence of false positives, reinforcing its suitability for macromolecules.

Two visualization strategies for macromolecules are presented, expanded and non-expanded, which directly influence how biotransformations are computed. The non-expanded mode reduces preprocessing time by minimizing the number of chemical structures that must be generated during analysis, with processing times ranging from 5 minutes for small peptides (<1000 Da) up to 40 minutes for larger peptides (>4000 Da). In contrast, the expanded mode simulates fragmentation at the atomic level and requires processing times ranging from 25 minutes for small peptides (<1000 Da) to several hours for larger peptides. For peptides larger than 4000 Da, the expanded mode becomes impractical due to excessively long processing times and high memory requirements. Moreover, both strategies can be combined in a hybrid approach, allowing selective expansion of specific monomers while keeping others non-expanded, as illustrated in the oligonucleotide dataset. This flexibility enhances interpretability by enabling targeted bond-level investigation of biotransformations without incurring the computational cost of expanding all the monomers within the compound.

## Supporting information

**S1 File. Metabolite identification reports exported from WebMetabase for each compound incubated in each protease using both algorithms.**
(PDF)

**S2 File. Dataset-1 and Dataset-2 MaM Settings.**
(PDF)

**S3 File. Dataset-1 and Dataset-2 MiM Settings.**
(PDF)

**S4 File. Dataset-2 Semaglutide MaM Settings.**
(PDF)

**S5 File. Dataset-2 Semaglutide MiM Settings.**
(PDF)

**S6 File. Dataset-3 MaM Settings DDA.**
(PDF)

**S7 File. Dataset-3 MiM Settings DDA.**
(PDF)

**S8 File. Dataset-3 MaM Settings DIA.**
(PDF)

**S9 File. Dataset-3 MiM Settings DIA.**
(PDF)

**S10 File. Dataset-4 MaM Settings.**
(PDF)

**S11 File. Dataset-4 MiM Settings.**
(PDF)

**S12 File. Dataset-5 MaM Settings.**
(PDF)

**S13 File. Dataset-6 MaM Settings.**
(PDF)

**S14 File. Dataset-6 MiM Settings.**
(PDF)

## Author contributions

**Data curation:** Paula Cifuentes López.

**Investigation:** Paula Cifuentes López.

**Methodology:** Paula Cifuentes López.

**Software:** Fabien Fontaine, Albert Garriga, Luca Morettoni.

**Supervision:** Ismael Zamora.

**Writing – original draft:** Paula Cifuentes López.

**Writing – review & editing:** Ismael Zamora, Tatiana Radchenko, Fabien Fontaine, Jesper Kammersgaard Christensen, Hans Helleberg, Bridget A. Becker.

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
