## [Decision Letter · Decision Letter 0]

10 Jun 2025

PONE-D-25-21950An automated software-assisted approach for exploring metabolic susceptibility and degradation products in macromolecules using High-Resolution Mass SpectrometryPLOS ONE

Dear Dr. Cifuentes López,

Thank you for submitting your manuscript to PLOS ONE. After careful consideration, we feel that it has merit but does not fully meet PLOS ONE’s publication criteria as it currently stands. Therefore, we invite you to submit a revised version of the manuscript that addresses the points raised during the review process. Both the reviewers have raised critical questions regarding use of ambiguous statements such as less time. Authors need to review the manuscript thoroghly to confirm complience with standard scientific language supported by citations or facts. There is a need for concrete description regarding Novelty as authors have similar works previously published. 

We look forward to receiving your revised manuscript.

Kind regards,

Yash Gupta, Ph.D.

Academic Editor

PLOS ONE

Journal Requirements:

3. Thank you for stating in your Funding Statement: [This work has been partially supported by Doctorats Industrials, AGAUR, Generalitat de Catalunya. Industrial Doctorate grant 00002/2023.].

4. Thank you for stating the following in the Competing Interests section: [The author declare the following competing financial interest(s): P.C. is an employee of Lead Molecular Design, S.L., and I.Z. is the CEO of the company. Lead Molecular Design, S.L. develops analytical software, including MassMetaSite and Oniro, which were used in this study.].

We note that you received funding from a commercial source: [Lead Molecular Design]

5. Thank you for stating the following in the Acknowledgments Section of your manuscript: [This work was supported by the Generalitat de Catalunya and Lead Molecular Design S.L through the Industrial Doctorate grant 00002/2023.]

Please remove any funding-related text from the manuscript and let us know how you would like to update your Funding Statement. Currently, your Funding Statement reads as follows: [This work has been partially supported by Doctorats Industrials, AGAUR, Generalitat de Catalunya. Industrial Doctorate grant 00002/2023.].

Reviewers' comments:

Reviewer's Responses to Questions

**Comments to the Author**

1. Is the manuscript technically sound, and do the data support the conclusions?

Reviewer #1: Yes

Reviewer #2: Yes

2. Has the statistical analysis been performed appropriately and rigorously? 

Reviewer #1: Yes

Reviewer #2: No

3. Have the authors made all data underlying the findings in their manuscript fully available?

Reviewer #1: Yes

Reviewer #2: Yes

4. Is the manuscript presented in an intelligible fashion and written in standard English?

Reviewer #1: Yes

Reviewer #2: Yes

5. Review Comments to the Author

Reviewer #1: 1. The authors have provided previous work citation for their sotftwares [6,7], which suggests some automating data analysis already exist. It would strengthen the paper highlight the novelty of this work such as is it the integration of peak detection algorithms for macromolecules (monoisotopic vs most abundant), the visualization approach.

2. Authors mentioned two peak detection techniques (MiM vs. MaM) and using fragment match scoring. For big molecules, for instance, how were mass inaccuracies (ppm) managed? Were several charge states taken into account and deconvoluted?

3. The authors claim “substantial reductions in processing time” and “consistent identification of degradation products” compared to prior work, but these claims need quantitative support. For processing time, it would be helpful to present a table or bar chart showing the workflow’s runtime for each dataset and algorithm (MiM vs MaM), alongside the runtimes of previous methods (if available from literature or the authors’ earlier pipelines). The statement "processing time ranged from 5 minutes to 2 hours per experiment," for instance, is helpful, but it would be more better to show how the specific circumstances (eg, large pepitde took 2h) and the extent of the improvement. Likewise, does "consistent identification" imply that the same metabolites were discovered in each replicate and also does the result of degardation matches previous reports?

4. Does these new algorithms and workflow are available to other researchers. It will better to mentioned whether the software is open-source or can be accessed (and how).

5. Authors are suggested to go for full experimental validation (e.g. NMR or orthogonal MS for each metabolite),

to support strong claims in the Conclusion and Abstract (“accurate prediction of metabolite structures”, “proven the analysis”, “greater confidence”).

Minor Comments

Ensure consistency in heading capitalization. eg. “Materials and methods” and “Results and Discussion”. Authors are advised to follow the journal’s style and apply it throughout the manuscript.

In Tables, either replace “Nº of dataset” with “Dataset number” or simply “Dataset”. Avoid using “Nº” (the degree symbol) in formal tables or define the same in its caption.

Both the Abstract and Conclusion summarize similar findings, improve both.

For consistency, either write out “minutes” or “min” uniformly, authors have used both which is need to be improved.

Check for grammer and acronyms at first use.

Reviewer #2: Major comments:

1. Please rewrite the introduction, including citations, that provide a scientific basis for the claims made compared to the existing literature.

2. In the introduction section, authors should:

(a) briefly explain how metabolites contribute to therapeutic efficacy, toxicity, and drug-drug interactions.

(b) highlight the importance of metabolite profiling that leads to the development of an effective drug in the introduction.

(c) explain the adverse effects of metabolites, preventing adverse drug reactions, and guiding clinical efforts.

(d) expand tools for in silico models and in vitro assays, identifying drug metabolites. Advantages and limitations of these approaches regarding the present study.

(e) detail more clear explanation of hybrid approaches with an example. What specific features of these software solutions faster than existing tools in terms of capabilities and performance?

(f) Please highlight any scientific report citing unsuitability for the conventional approach for larger and intricate molecules such as peptide oligonucleotide and antibodies, with increasing challenges with the increase in size.

(g) Line 99:101 – The article aims to address the challenges in the processing of only macromolecules, or macromolecules and small molecules, as described in the manuscript?

(h) Line 107:111 – Datasets consisted of some linear and cyclic as well as natural and unnatural amino acids, and oligonucleotides, can represent the scope of application for analysis on a wide variety of compounds? How the coverage of applicability has been defined with compounds covered in the datasets. Please define

(i) Line 141:144 – Repeated lines.

3. Authors should provide an alternate approach when the fragmentation data does not match well with the theoretical.

4. How does this system handle data quality control and verification? Is it an entirely automated process? What threshold detail can it provide for the user to make an accurate decision?

5. These algorithms are based on fragmentation, isotope pattern, and m/z differences. Is there any possibility to combine these software with other data types for broad information, like 3D structure prediction? Also, quantitative details are required for more reliable weightage.

6. It would be great to mention the statistical analysis of the results throughout. Are these statistically significant? Also, mention these differences between the two algorithms.

7. For high score context, authors should mention a specific numerical threshold.

8. Have previous studies conducted similar studies or comparisons? Please cite them and mention your novelty.

9. MiM and MaM algorithms have the distinction of handling larger peptides. Elaborate underlying reason behind it. Different ways to process peptide size or mass could be a reason for it?

10. Please comment on how score differences reflect the practical reliability of the identification. Are the score differences substantial enough for metabolite identification? Also, mention scores for false-positive or false-negative identification. Authors should discuss the score threshold in all analysis.

11. Please comment: Is MiM less sensitive to certain types of enzyme degradation? OR the score differences related to the algorithm's sensitivity.

12. Comment on how these algorithms handle false positives.

13. You mention that the non-expanded structure visualization helped achieve the reduced processing time. Could you clarify why this approach is more efficient compared to the expanded visualization?

14. Why were particular proteases like cathepsin G, neutrophil elastase, and trypsin chosen?

15. Line 247:251 – Generations of metabolites are classified with virtual screening based on predefined biotransformation reactions and m/z ratio, but differences among metabolites due to stereochemistry and reactions outside of the considered biotransformation limit the chances of correct analysis. How do authors believe that their method has overcome these limitations?

16. Line 411: 415 – classification of contaminations and background noise by unrecognised and irregular peaks may correspond to variation among metabolites of different generations. How do authors justify their parameters outlining the peak selection?

17. Line 315:317 – Dependency on the predefined databases should be countered if the structure is not matched?

18. Could this approach be applied to high-throughput screening of peptide libraries or to analyse metabolites in clinical samples?

6. PLOS authors have the option to publish the peer review history of their article (what does this mean? ). If published, this will include your full peer review and any attached files.

**Do you want your identity to be public for this peer review?** For information about this choice, including consent withdrawal, please see our Privacy Policy .

Reviewer #1: **Yes: ** SUMIT KUMAR

Reviewer #2: No

---

## [Author Response · Author response to Decision Letter 1]

10 Jul 2025

Dear Editor and Reviewers,

We sincerely thank you for your thoughtful and constructive comments on our manuscript. We have carefully considered each point raised and have revised the manuscript accordingly to address all concerns. We provide detailed responses to the academic editor and reviewers’ comments along with explanations of the changes made in the "Response to Reviewers" document.

---

## [Decision Letter · Decision Letter 1]

30 Jul 2025

An automated software-assisted approach for exploring metabolic susceptibility and degradation products in macromolecules using High-Resolution Mass Spectrometry

PONE-D-25-21950R1

Dear Dr. Cifuentes López,

We’re pleased to inform you that your manuscript has been judged scientifically suitable for publication and will be formally accepted for publication once it meets all outstanding technical requirements.

Kind regards,

Yash Gupta, Ph.D.

Academic Editor

PLOS ONE

Additional Editor Comments (optional):

Reviewers' comments:

Reviewer's Responses to Questions

**Comments to the Author**

1. If the authors have adequately addressed your comments raised in a previous round of review and you feel that this manuscript is now acceptable for publication, you may indicate that here to bypass the “Comments to the Author” section, enter your conflict of interest statement in the “Confidential to Editor” section, and submit your "Accept" recommendation.

Reviewer #1: All comments have been addressed

Reviewer #2: All comments have been addressed

2. Is the manuscript technically sound, and do the data support the conclusions?

Reviewer #1: Partly

Reviewer #2: Yes

3. Has the statistical analysis been performed appropriately and rigorously? 

Reviewer #1: Yes

Reviewer #2: Yes

4. Have the authors made all data underlying the findings in their manuscript fully available?

Reviewer #1: Yes

Reviewer #2: Yes

5. Is the manuscript presented in an intelligible fashion and written in standard English?

Reviewer #1: Yes

Reviewer #2: Yes

6. Review Comments to the Author

Reviewer #1: Ajuthors have addressed the point raised, and manuscript can be accepted for publication in the journal.

Reviewer #2: (No Response)

7. PLOS authors have the option to publish the peer review history of their article (what does this mean? ). If published, this will include your full peer review and any attached files.

**Do you want your identity to be public for this peer review?** For information about this choice, including consent withdrawal, please see our Privacy Policy .

Reviewer #1: **Yes: ** Sumit Kumar

Reviewer #2: **Yes: ** Bhawna Saini

---

## [Editor Report · Acceptance letter]

PONE-D-25-21950R1

PLOS ONE

Dear Dr. Cifuentes López,

I'm pleased to inform you that your manuscript has been deemed suitable for publication in PLOS ONE. Congratulations! Your manuscript is now being handed over to our production team.

Kind regards,

on behalf of

Dr. Yash Gupta

Academic Editor

PLOS ONE